# Properties of Arctic liquid and mixed phase clouds from ship-borne Cloudnet observations during ACSE 2014

Peggy Achtert[1,2], Ewan J. O'Connor[3,4], Ian M. Brooks[1], Georgia Sotiropoulou[5,6], Matthew D. Shupe[7,8], Bernhard Pospichal[9], Barbara J. Brooks[10], and Michael Tjernström[5]

[1]Institute for Climate and Atmospheric Science, School of Earth and Environment, University of Leeds, Leeds, UK
[2]Now at Meteorological Observatory Hohenpeißenberg, German Weather Service, Germany
[3]Finnish Meteorological Institute, Helsinki, Finland
[4]Meteorology Department, University of Reading, Reading, UK
[5]Department of Meteorology and the Bert Bolin Centre for Climate Research, Stockholm University, Stockholm, Sweden
[6]Now at Laboratory of Atmospheric Processes and Their Impacts, School of Architecture, Civil & Environmental Engineering, École Polytechnique Fédérale de Lausanne (EPFL), Lausanne, Switzerland
[7]Cooperative Institute for the Research in Environmental Sciences, University of Colorado Boulder, Boulder, Colorado, USA
[8]Earth System Research Laboratory, National Oceanic and Atmospheric Administration, Boulder, Colorado, USA
[9]Institute for Geophysics and Meteorology, University of Cologne, Cologne, Germany
[10]National Centre for Atmospheric Science, University of Leeds, Leeds, UK

**Correspondence:** Peggy Achtert (p.tesche-achtert@dwd.de)

**Abstract.** This study presents Cloudnet retrievals of Arctic clouds from measurements conducted during a three-month research expedition along the Siberian shelf during summer and autumn 2014. During autumn, we find a strong reduction in the occurrence of liquid clouds and an increase for both mixed-phase and ice clouds at low levels compared to summer. About 80% of all liquid clouds observed during the research cruise show a liquid water path below the infra-red black body limit of approximately $50\,\mathrm{gm}^{-2}$. The majority of mixed-phase and ice clouds had an ice water path below $20\,\mathrm{gm}^{-2}$.

Cloud properties are analysed with respect to cloud-top temperature and boundary layer structure. Changes in these parameters have little effect on the geometric thickness of liquid clouds while mixed-phase clouds during warm-air advection events are generally thinner than when such events were absent. Cloud-top temperatures are very similar for all mixed-phase clouds. However, more cases of lower cloud-top temperature were observed in the absence of warm-air advection.

Profiles of liquid and ice water content are normalised with respect to cloud base and height. For liquid water clouds, the liquid water content profile reveals a strong increase with height with a maximum within the upper quarter of the clouds followed by a sharp decrease towards cloud top. Liquid water content is lowest for clouds observed below an inversion during warm-air advection events. Most mixed-phase clouds show a liquid water content profile with a very similar shape to that of liquid clouds but with lower maximum values during events with warm air above the planetary boundary layer. The normalised ice water content profiles in mixed-phase clouds look different from that of liquid water content. They show a wider range in maximum values with lowest ice water content for clouds below an inversion and highest values for clouds above or extending through an inversion. The ice water content profile generally peaks at a height below the peak in the liquid water content profile – usually in the centre of the cloud, sometimes closer to cloud base, likely due to particle sublimation as the crystals fall through the cloud.

## 1  Introduction

Over the past 30 years the rate of Arctic warming has been consistently larger than the global average, by a factor of 2-3 (IPCC, 2013). This has led to a decrease in sea-ice cover and new record minima in the late summer sea-ice extent in the Arctic occurred in 2007 and 2012. The warming of the Arctic prolongs the sea-ice melt season (Markus et al., 2009), which specifically reduces

the cover of perennial sea ice (Maslanik et al., 2011). There is not yet a consensus regarding which mechanisms dominate the rapid warming in the Arctic. Although climate models agree on an enhanced Arctic warming, sometimes referred to as the Arctic amplification (Polyakov et al., 2002; Serreze and Francis, 2006; Serreze and Barry, 2011), they largely fail to predict the accelerated retreat of Arctic sea ice (Stroeve et al., 2012). This is at least partly caused by an inadequate description of the processes that control the coupled oceanic-atmospheric energy balance and the feedback mechanisms between sea-ice cover

and other components of the Arctic climate system (Liu et al., 2012a), particularly clouds.

Arctic low- and mid-level clouds can differ significantly from their counterparts at lower latitudes. They are generally long-lived and of mixed-phase nature (Shupe, 2011) whose macrophysical (base and top altitudes, horizontal extent), microphysical properties (e.g., cloud droplet and ice crystal number concentrations, liquid water path (LWP), ice water path (IWP), and liquid-ice partitioning) and radiative effects are influenced by the low aerosol particle – cloud condensation nuclei (CCN) and

ice nucleating particle (INP)– number concentrations during summer (Mauritsen et al., 2011; Birch et al., 2012; Tjernström et al., 2014; Hines and Bromwich, 2017). The aerosol particle size distribution can affect the distributions of, and the feedback between, liquid water and ice particles in the clouds, and thus impact the radiative properties of the clouds (Solomon et al., 2009). In addition, temperature and moisture inversions influence entrainment at cloud top with consequences for cloud development (Sedlar and Tjernström, 2009; Sedlar et al., 2012; Solomon et al., 2011).

The impact of Arctic clouds on solar and terrestrial radiation is not well quantified, and hence the accurate description of the atmospheric and surface energy budgets remains one of the core problems in Arctic climate modelling (Karlsson and Svensson, 2011; Boeke and Taylor, 2016). Low-level liquid-water and mixed-phased clouds are of particular importance, typically evolving through cloud-top radiative cooling and consequent turbulent mixing and entrainment of warm and humid air. They form in statically stable atmospheric conditions, and persist for extended periods of time. Steele et al. (2010) show

that about 60% of the energy that is consumed by the melting sea ice during the melting season is provided by radiative energy or sensible heat fluxes directly from the atmosphere to the surface, both strongly modified by clouds. Hence, even small errors in parameters affecting the downward radiative fluxes absorbed and emitted by clouds, such as cloud cover, microphysical, macrophysical and optical properties (Tjernström et al., 2008; Walsh et al., 2009; Birch et al., 2009, 2012; Hines and Bromwich, 2017), may have far-reaching consequences on the surface energy budget in the Arctic (Sedlar et al.,

2011; Bennartz et al., 2013; Ebell et al., 2020), and consequently on ice melt (Tjernström, 2005).

Of particular importance is the thermodynamic phase of the clouds in the Arctic as it significantly affects their radiative effect (Shupe and Intrieri, 2004; Choi et al., 2014; Komurcu et al., 2014; Tan et al., 2016). For instance, the widespread occurrence of

warm liquid-water clouds, i.e. clouds with top temperature above 0°C, as identified in remote-sensing observations collected during the Arctic Clouds in Summer Experiment (ACSE) has been associated with observations of rapid decrease in sea-ice cover (Tjernström et al., 2015). A complicating factor is that the properties and behaviour of Arctic boundary-layer clouds may differ with region. For example, a statistical analysis of radiative properties of the clouds observed during ACSE showed that knowledge derived from measurements across the pan-Arctic area and on the central ice-pack does not necessarily apply closer to the ice-edge (Sotiropoulou et al., 2016). In addition, cloudiness and its effect on the energy balance at the surface strongly depends on the change in specific humidity within surface inversions (Tjernström et al., 2019).

This paper continues the investigation of the clouds observed during the ACSE expedition, focussing on their properties as derived from synergetic remote-sensing measurements. Such information is needed to improve the understanding necessary to improve representation of Arctic clouds in global numerical weather prediction and climate models.

## 2 Measurements and methods

### 2.1 The field campaign

ACSE was part of the Swedish-Russian-US Arctic Ocean Investigation of Climate-Cryosphere-Carbon Interactions (SWERUS-C3) project. Measurements were made during a 3-month research cruise on the icebreaker Oden, from 3 July to 5 October 2014. The expedition started from Tromsø, Norway, and followed the Siberian Shelf, crossing the Kara, Laptev, East Siberian, and Chukchi Seas to arrive off Barrow, Alaska, on 19 August. Following a change of crew and science teams, Oden returned to Tromsøon a route somewhat to the north of the outbound leg. The cruise track is shown in Figure 1 together with the tracks of research cruises undertaken in two previous projects: the Surface Heat Budget of the Arctic Ocean (SHEBA, Uttal et al. (2002)) and Arctic Summer Cloud Ocean Study (ASCOS, Tjernström et al. (2014)) experiments. One of the primary aims of ACSE was to investigate the effect of different surface conditions (i.e., open water, marginal ice zones, and sea ice) on the macrophysical and microphysical properties of Arctic low- and mid-level clouds through the late summer melt season into the early autumn freeze up.

In contrast to the majority of shipborne cloud observations in the Arctic, ACSE measurements were performed when the ship was moving. Hence, the measurements were taken over open water as well as partial or complete sea ice cover, and the ice appeared with and without melt ponds and snow cover. The moving platform complicates a statistical analysis of the meteorological situation and we provide only a basic overview here. Meteorological instruments and the measured conditions are presented in more detail in Sotiropoulou et al. (2016) and Sedlar et al. (2020). The impact of meridional heat transport on the surface energy budget during ACSE is described in Tjernström et al. (2015, 2019).

Sotiropoulou et al. (2016) used the lower atmospheric thermal structure as inferred from 6-hourly soundings to separate the ACSE period into two seasons (see their Figure 2). Before 1200 UTC on 27 August 2014, relatively high temperatures prevailed in the lower troposphere up to 5 km height, with occasional cooler periods. Several strong warm-air advection events occurred during this first half of the experiment, which Sotiropoulou et al. (2016) refer to as the summer melt season. The strongest occurred in the beginning of August 2014 and is described in Tjernström et al. (2015). After 27 August 2014 the lowermost

kilometres of the atmosphere changed substantially with a decreased temperature, with only a few occasional warmer events, considered to represent autumn freeze-up (about 42% of the ACSE time period). Figure 2b in Sotiropoulou et al. (2016) shows the temperature at the main inversion, i.e. the strongest inversion in the radiosonde profiles used as a proxy for the top of the boundary layer. This is mostly positive for the summer period and generally negative during the autumn period.

Figure 2 in Sedlar et al. (2020) present time series of select near-surface meteorological parameters, indicating a number of synoptic weather events that were encountered along the ACSE track. These events occurred more often in the second half of the experiment, though with a shorter duration. Surface pressure minima first dropped below 1,000 hPa on 27 August, the same date that Sotiropoulou et al. (2016) defined as the seasonal transition from summer to autumn. Near-surface wind speed also peaks more often and slightly higher after this date, compared to earlier. The transition is also visible in near-surface

temperature, which remained at or above freezing level before 27 August then fell down to, or below, the freezing point. Figure 2 shows that fog occurred much more frequently during the summer melt season compared to the autumn freeze-up. The difference in cloud occurrence and properties between the two seasons is discussed in detail later in this paper.

## 2.2 Instrumentation and data processing

The suite of remote-sensing instruments employed in this study comprise a W-band Doppler cloud radar (National Oceanic and

Atmospheric Administration, Boulder, USA), a motion-corrected Doppler wind lidar (HALO Photonics, Achtert et al. (2015)), a laser ceilometer (Vaisala CL31), and a scanning microwave radiometer (Radiometer Physics HATPRO). The W-band cloud radar is a motion-stabilised system developed specifically for shipborne deployments (Moran et al., 2012) operating at 94 GHz and measuring the Doppler spectrum from which the first three moments (reflectivity, Doppler velocity, Doppler spectrum width) are calculated. It is a pulsed system and provides vertical profiles with 31.22 m vertical resolution and 0.5 s temporal

resolution. During ACSE, the lowest and highest range gates were set to 80 m and 5980 m, respectively.

The Doppler lidar is a pulsed heterodyne system operating at a wavelength of 1.5 $\mu$m and a pulse repetition frequency of 15 kHz. Range resolution was set at 18 m and 30 000 pulses accumulated to achieve a temporal resolution of 2 s. The scan schedule comprised a fixed schedule for the entire voyage of a continuous vertical stare mode interspersed with a five-beam wind scan every 10 minutes at an elevation angle of 70 degrees from horizontal. A full description of the system parameters

and scan schedule is given in Achtert et al. (2015). The Doppler lidar signal was corrected following Manninen et al. (2016). This new background correction, developed for measurements in low-aerosol conditions, improves the signal to noise ratio threshold for reliable data by about 4 dB above the original signal threshold (Achtert et al., 2015), increasing data availability and providing more reliable Doppler velocity uncertainty estimates.

The ceilometer operates at a wavelength of 905 nm with a vertical resolution of 10 m. Pulses are accumulated to a temporal

resolution of 30 s. The instrument was deployed pointing to zenith.

The microwave radiometer is a RPG-HATPRO-G1, which is a passive system monitoring 14 channels in two frequency bands (7 for humidity profiling and liquid water path retrievals between 22 and 31 GHz; 7 for temperature profiling between 51 and 58 GHz). We retrieve the liquid water path (LWP) from the raw microwave brightness temperature measurements following Löhnert and Crewell (2003) and Massaro et al. (2015). This statistical retrieval requires climatological profiles of

pressure, temperature and humidity as derived from 6-hourly soundings. A suitable training data set was assembled from a total of 1826 radiosondes launched in the Arctic Ocean from the research vessels Polarstern (https://data.awi.de/?site=home), Mirai (http://www.godac.jamstec.go.jp/darwin/e), and Oden (https://bolin.su.se/data/) between 1990 and 2014. This includes the soundings performed during ACSE. The soundings were separated according to summer (June, July, August, 1025 radiosondes) and autumn (September, October, 801 radiosondes). LWP measurements are limited to non-precipitating cases as heavy rain can impact the LWP retrieval (Crewell and Löhnert, 2003).

Liquid clouds are diagnosed from lidar and radar profiles. The microwave radiometer provides the LWP associated with these clouds. The resolution of the LWP retrieval is about $5\,\mathrm{gm}^{-2}$, but the uncertainty in LWP is still of the order of 20-$30\,\mathrm{gm}^{-2}$ (Turner, 2007). An offset correction is done by using the lidar to diagnose clear sky periods when LWP should be zero and adjust the coefficients for the microwave radiometer retrieval to obtain values around zero. Details are provided in (Gaussiat et al., 2007). A similar approach is also used for the Atmospheric Radiation Measurement (ARM) MWR RETrieval (MWRRET) retrieval (Turner et al., 2007). This offset correction leads to a bias that is much lower than $20\,\mathrm{gm}^{-2}$ Values of LWP below $\approx25\,\mathrm{gm}^{-2}$ in the presence of clouds (as detected from independent measurements) and a known bias (i.e. from offset correction during clear-sky periods) must not be tossed as this leads to a bias in the statistics. Those are still valid values though with an error of $\pm\approx25\,\mathrm{gm}^{-2}$.

A time series of LWP and the raw measurements of brightness temperature at 31.4 GHz as observed at relatively low LWP values during a change from cloudy to cloud-free conditions for a single-layer cloud is shown in Figure 2. The brightness temperature varies between 17°C and 18°C when the retrieval gives a LWP around $10\,\mathrm{gm}^{-2}$ during the time period from 1100 to 1200 UTC on 25 July 2014. Visibility is very low until 1400 UTC and the Doppler lidar signal appears to be fully attenuated by the cloud. The cloud produced some precipitation from 1330 UTC and started to disappear around 1400 UTC, when the lidar was able to fully penetrate the cloud. The same data is shown in Figure 3 as a scatter plot of LWP and brightness temperature colour coded according to the maximum in radar reflectivity of the respective profile. The figure shows that the 31.4 GHz brightness temperature is lowest at LWP around $0\,\mathrm{gm}^{-2}$, i.e. in the absence of cloud, supporting the plausibility of the LWP retrieval and the offset correction. Further example cases of a cloud system that extends through the inversion and of a cloud system that is precipitating ice versus liquid are shown in the Supplementary Material.

Surface meteorology measurements included air temperature, humidity, mean and turbulent winds, visibility, and downwelling solar and infra-red radiation (Sedlar et al., 2020). Radiosondes (Vaisala RS92) were launched four times a day at 0000, 0600, 1200, and 1800 UTC (Sotiropoulou et al., 2016).

These measurements allow for a comprehensive characterisation of clouds using the Cloudnet algorithm (Illingworth et al., 2007), combining cloud radar, ceilometer, microwave radiometer and radiosonde profiles averaged to a common grid at the cloud radar resolution. The radiosonde profiles provide the initial temperature and humidity profiles for Cloudnet. They also supply the information necessary to estimate and correct for gaseous and liquid attenuation of the radar reflectivity. Gaseous attenuation at 94 GHz is not so severe in Arctic conditions but may reach 1 dB already within 2 km, whereas attenuation by liquid cloud layers can reach 2 dB or more. This attenuation, if uncorrected for, would cause a significant bias in derived ice water contents (IWC), especially if occurring above liquid layers. Together with the re-gridded remote-sensing data, the

Cloudnet scheme also provides an objective hydrometeor target classification at the same cloud radar resolution; the re-gridded data and the target classification are combined in a single file termed the target categorisation product which also contains the measurement uncertainties for propagation through to all products. The measurement uncertainties of the individual instruments are used to obtain a data quality flag. This study only considers profiles that are flagged as reliable and show a standard deviation of the LWP smaller than 120 $gm^{-2}$.

The target categorisation product is the basis for deriving consistent retrievals of cloud occurrence, top and base height, cloud thickness, cloud phase, liquid and ice-water path, liquid and ice water content, and the effective radius of cloud droplets and ice crystals. Liquid water content (LWC) is calculated from microwave radiometer-derived LWP (with an offset correction based on clear-sky periods) and liquid layer cloud boundaries by distributing the liquid using the scaled-linear adiabatic assumption, i.e. LWC increasing linearly with height from zero at cloud base (Albrecht et al., 1990; Boers et al., 2000). Typical errors in LWC are below 20% (Ebell et al., 2010). IWC is calculated from radar reflectivity and temperature using the method of Hogan et al. (2006), where the fractional error in IWC at 94 GHz is +55%/-35% between -10 and -20 C, rising to +90%/-47% for temperatures below -40°C. Note that an error in the calibration of the radar reflectivity of 1 dB would bias IWC by 15%.

The Cloudnet target classification Illingworth et al. (2007) has been used to separate between water clouds, ice clouds, and mixed-phase clouds on a profile-by-profile basis with a resolution of 30 s, and to identify cloud base and top heights. The original Cloudnet target classification for the three months of ACSE measurements is presented in Figure 4. The figure also shows fog periods as identified by a visibility of less than 1 km in the 10-min mean of the visibility sensor measurements aboard Oden. The target classification reveals an unrealistically high occurrence of *Aerosol*, *Aerosol & insects*, and *Insects* during periods that were actually dominated by fog. Hence, visibility data have been used to re-classify some of the targets originally misidentified by Cloudnet into these categories below 500 m as fog. A cloud is defined as liquid if its profile contains only height bins that are classified as *Cloud droplets only* or *Drizzle/rain & cloud droplets*. A cloud for which all height bins are classified as *Ice only* is defined as ice cloud. A cloud layer is defined as mixed-phase, if it contains any possible combination of *Ice only*, *Cloud droplets only*, *Melting ice*, *Melting ice & cloud droplets*, and *Ice & super-cooled droplets*. Finally, layers of *Cloud droplets only* with precipitating ice below cloud base and mixed-phase clouds with *Drizzle or rain* below cloud base are defined as precipitating mixed phase. Liquid clouds with liquid precipitation are defined as precipitating liquid. Profiles of cloud fraction per volume (Brooks et al., 2005) have been obtained using time-height sections of 30 min by 90 m height (3 height bins). When comparing our findings to results from previous studies that use the cloud classification of Shupe (2007), we sort all layers of *Cloud droplets only* with precipitating ice (*Ice only*) below cloud base into the mixed-phase category to be in line with the earlier work.

We use the estimates of the depth of the planetary boundary layer (PBL) provided by Sotiropoulou et al. (2016). They obtained PBL depths from the locations of the main inversions in the radiosonde temperature profiles following the methodology of Tjernström and Graverson (2009). A separation between coupled and decoupled boundary layers (Shupe et al., 2013; Sotiropoulou et al., 2014; Brooks et al., 2017) was performed by investigating the presence of an additional, weaker, temperature inversion below the main inversion (Sotiropoulou et al., 2016). An absence of such an additional lower inversion defines coupled PBLs. Cloudnet retrievals within one hour of a sounding have been used in the investigation of the effects of (a) cou-

pled and decoupled PBLs and (b) the location of the clouds with respect to the inversion (i.e. PBL top) on the observed cloud properties. To avoid oversampling of persistent clouds, we considered only one Cloudnet profile every 5 minutes, leading to at most 24 profiles for per sounding.

Based on sounding data taken during ACSE, Sotiropoulou et al. (2016) defined the change between summer and autumn by a rapid change in temperature in the lower atmosphere on 28 August 2014. Here, we use this date to investigate changes in the observed cloud properties and occurrence rates between the two seasons. We further separate between conditions during which warm air was present at 1 km height, i.e. in the free troposphere, (warm air events, WAE) and during which it was not (non-warm air events, non-WAE). WAE were identified from the ACSE soundings as when the temperature at 1.0 km height exceeded a threshold of 5°C, empirically derived from Figure 2a of Sotiropoulou et al. (2016). These events were particularly pronounced during the ACSE summer observations and are likely the result of warm-air advection from lower latitudes (Tjernström et al., 2015, 2019).

The investigation of clouds in this study is restricted to heights below 6 km, the maximum height of the cloud radar observations during ACSE. For the statistical analysis of the occurrence of different cloud types and cloud layers, we hence include only those clouds that show a cloud-top height below 6 km, considering up to three cloud layers per profile. This means that deep mid-level clouds and cirrus are not fully covered in our data set.

# 3 Results

## 3.1 Cloud occurrence

Cloud occurrence probability distributions as a function of height are shown in Figure 5, both for total occurrence and partitioned into liquid, precipitating liquid, ice, mixed-phase, and precipitating mixed-phase clouds for the entire ACSE campaign, and separated into summer and the autumn seasons following Sotiropoulou et al. (2016). For completeness, the cloud fraction for all clouds, i.e. including those with a cloud-top height above 6 km for which only cloud base could be detected, is provided as dotted line.

In general, Figure 5 shows that clouds were more abundant below 4 km height during autumn than during summer. This is reflected in the lower tropospheric maxima of the mean cloud fraction of 0.42 and 0.75 in summer and autumn, respectively. In summer, there is a clear separation between height ranges dominated by liquid-water (< 1.2 km) and by ice clouds (> 1.2 km). Precipitating and non-precipitating mixed-phase clouds during summer were found at all height levels though their cloud fraction strongly decreased upwards of 0.5 km. Autumn showed a strong reduction in the occurrence of liquid clouds and an increase in both mixed-phase clouds and ice clouds at low levels. Ice clouds during autumn extended almost down to the surface, while low clouds during summer were predominantly liquid. Figure 5 also reveals that precipitating clouds were more abundant during summer than during winter. This is in line with Figure 4 which shows that precipitating clouds are linked to frontal passages, i.e. deep cloud systems. More of such deep cloud systems have been encountered during summer. In contrast, a stable boundary layer with shallow stratus clouds, which typical occur in the marginal ice zone, prevailed during autumn.

Only remote-sensing observations and the Cloudnet target classification are used to identify cloud layers. This gives apparent cloud layers and apparent multi-layer clouds for which features have to be clearly separated in a profile. During summer, this approach gives occurrence rates of 19.6% cloud-free conditions, 39.1% single-layer clouds, and 41.3% multi-layer clouds. During autumn, these numbers change to 4.6%, 47.6%, and 47.8%. This means that apparent single-layer clouds and multi-layer clouds occur at about the same rate during both seasons.

A statistical overview of top temperature, top height, bottom height, and geometrical thickness of the clouds observed during ACSE is provided in Figure 6. The results refer to cloud layers (up to three allowed per profile) for which both cloud base and top could be clearly identified. The minimum cloud geometrical depth was defined by the radar range resolution of 31 m. Again, the results were separated according to cloud phase and season. Average cloud top temperatures were 0°C for liquid clouds, -10°C for mixed-phase clouds, and -15°C for ice clouds. Cloud top temperatures were slightly higher during summer and for precipitating clouds and slightly lower during winter and for non-precipitating clouds, though with a similar spread of values. The seasonal behaviour of cloud top and base heights for liquid clouds differs from that of ice and mixed-phase clouds. Liquid clouds were relatively unchanged in vertical extent between summer and autumn, while both ice and mixed-phase clouds had lower top and base heights in autumn than in summer. Note that the increased top height and cloud thickness of precipitating mixed-phase clouds is related to their connection to the passage of low pressure systems.

In general, the clouds observed during ACSE were rather shallow with a median (mean) geometrical thickness of 250 m (800 m). Liquid clouds were found to be thinnest during both seasons and with only a small variation between median (220 m) and mean (285 m) values. Mixed-phase clouds were the thickest with median depths of 750 m in summer and 940 m in winter, with a similar mean value for both seasons. Ice clouds were slightly deeper in autumn, with a median (mean) geometric thickness of 250 m (730 m) compared to 220 m (570 m) in summer. It should be emphasised that these statistics are dominated by liquid clouds in summer and by mixed-phase clouds during autumn.

## 3.2 LWP, IWP and cloud top temperature

### 3.2.1 Liquid-water clouds

The frequency distribution of LWP in non-precipitating liquid water clouds during summer and autumn is shown in Figure 7a. While a negative LWP related to the retrieval error of $25\text{-}30\,\mathrm{gm}^{-2}$ (Turner et al., 2007) is clearly unphysical, these values cannot be excluded without biasing the statistics. Liquid water clouds during summer had a mean LWP of $37\pm59\,\mathrm{gm}^{-2}$ and median of $13\,\mathrm{gm}^{-2}$. These values were similar during autumn with a mean of $41\pm54\,\mathrm{gm}^{-2}$ and median of 20 g/m2. Both distributions peak at a LWP around $10\,\mathrm{gm}^{-2}$. In summer a small number of clouds (less than 1% of all cases) had a LWP in excess of $400\,\mathrm{gm}^{-2}$ while in autumn the maximum LWP was approximately $495\,\mathrm{gm}^{-2}$. These high values of LWP are generally related to frontal passages. Almost no seasonal difference in the LWP distributions is apparent in the cumulative frequency curves in Figure 5a. The curves also show that in summer and autumn 76% and 72%, respectively, of liquid clouds were below the infra-red black body limit of approximately $50\,\mathrm{gm}^{-2}$ (Tjernström et al., 2015). If the black body limit is set to $30\,\mathrm{gm}^{-2}$ (Shupe and Intrieri, 2004), the occurrence rates are reduced to about 67% in summer and 60% in autumn. These

clouds were therefore often semi-transparent to long-wave radiation; hence, long-wave cooling and the resulting turbulence generated in cloud, as well as the surface downwelling radiation, will be very sensitive to small changes in LWP.

Figure 7b shows the distribution of cloud-top temperature for liquid-water clouds during summer and autumn. Summer liquid clouds were warmer than those in winter. Their cloud top could be warmer than 15°C but were never found to be colder than -15°C. A closer look at the data revealed that all the cloud-top temperatures above 10°C were the result of a period of strong warm air advection that occurred in the beginning of August (Tjernström et al., 2015, 2019). The cloud-top temperature distribution observed during summer resembles that derived from Cloudnet observations at mid-latitudes (Bühl et al., 2016). In autumn, liquid cloud-top temperatures rarely exceed 0°C with observed values as low as -25°C. The maximum of cloud-top temperature occurrence rate shifts from 0°C in summer to -5°C in autumn. In addition, cloud-top temperatures for autumn also show a broader distribution with a long tail towards low temperatures than those in summer.

### 3.2.2 Mixed-phase clouds

The LWP frequency distribution for mixed-phase clouds (including liquid clouds with ice below) presented in Figure 8a is similar to that for liquid-only clouds in Figure 7a though with a broader shape. Summer had more cases of high LWP and fewer cases of low LWP than autumn. For both seasons, the peak occurrence was at around $10\,\mathrm{g\,m^{-2}}$. The mean and median values, however, are higher than for liquid-only clouds, with summer values of $98\pm94\,\mathrm{g\,m^{-2}}$ and $72\,\mathrm{g\,m^{-2}}$, respectively; in autumn the corresponding values are $34\pm44\,\mathrm{g\,m^{-2}}$ and $21\,\mathrm{g\,m^{-2}}$. The cumulative distributions in Figure 8a show that, with infrared-black body limit of $50\,\mathrm{g\,m^{-2}}$ ($30\,\mathrm{g\,m^{-2}}$), 41% (31%) and 76% (60%) of the clouds during summer and autumn, respectively, had LWPs below this limit. The same general relationships of higher median LWP in mixed-phase clouds compared with liquid-only clouds is consistent with the observations during SHEBA (Shupe et al., 2006).

In contrast to LWP, there is little difference in the frequency distributions for IWP in the mixed-phase clouds observed in either summer or autumn (Figure 8b). The majority of clouds had an IWP below $20\,\mathrm{g\,m^{-2}}$ with mean and median values in summer of 34 and $7\,\mathrm{g\,m^{-2}}$, respectively, and in autumn of 32 and $9\,\mathrm{g\,m^{-2}}$.

During summer, IWC was lowest in clouds with a low cloud top height and highest for clouds with tops between 3.0 and 4.0 km and cloud-top temperatures of -8°C to -17°C (not shown). During autumn, the lowest values of IWC were observed for clouds with top heights in the range from 2.0 to 3.0 km. Cold clouds with cloud top temperatures between -15°C and -35°C and cloud top heights above 4.0 km had the largest values of IWC (not shown). The majority of mixed-phase clouds during summer and autumn had very low IWC; $< 0.1\,\mathrm{g\,m^{-3}}$. Mean (median) values were $0.0156\,\mathrm{g\,m^{-3}}$ ($0.0025\,\mathrm{g\,m^{-3}}$) and $0.0087\,\mathrm{g\,m^{-3}}$ ($0.0016\,\mathrm{g\,m^{-3}}$) during summer and autumn, respectively.

The frequency distribution of cloud-top temperature in Figure 8c again shows a different behaviour for clouds during summer and autumn. During summer, the tops of mixed-phase clouds were generally warmer than in autumn with a maximum at 0°C to -2.5°C. However, they were always colder than liquid-only clouds during the same season. During summer, cloud-top temperature could be as low as -30°C though they were mostly warmer than -5°C. Autumn had a bi-modal distribution of cloud-top temperature, which could be the result of precipitating ($T_{\mathrm{top}}$ >-10°C) versus non-precipitating clouds ($T_{\mathrm{top}}$ <-10°C) (Westbrook and Illingworth, 2011). Very few mixed-phase clouds showed cloud-top temperatures above 0°C (these were cases

related to warm-air advection events where the cloud top extended into the warmer air above) or as low as -35°C. In general, mixed-phase cloud top temperatures were up to 5°C colder during autumn than during summer.

### 3.2.3 Effect of boundary-layer structure

Here we investigate the effects of PBL structure on the observed clouds. The PBL top is defined as the height of the strongest temperature inversion (Brooks et al., 2017) within the lowermost 3 km of the atmosphere (Sotiropoulou et al., 2016). Clouds are considered to be *below* the inversion (cloud top below the PBL top), *above* the inversion (cloud base above the PBL top), or to *extend into* the inversion (cloud base below PBL top and cloud top above PBL top).

Figures 9 and 10 provide a statistical overview of the geometrical thickness and cloud-top temperature of clouds for different PBL structure and temperature at 1 km height. We separate between liquid and mixed-phase clouds observed above, below, and extending into the inversion during WAE and non-WAE conditions as well as for coupled and decoupled PBLs. Cases of liquid and mixed-phase clouds in decoupled PBLs during WAE were rare (N<100) in the ACSE data set, and thus, not considered here. Non-precipitating and precipitating liquid clouds showed little difference in mean and median cloud thickness (Figures 9a and c). However, they do show a clear difference in the frequency distribution of cloud-top temperature with respect to WAE and non-WAE conditions and whether or not they are precipitating (Figures 9b and d). Cloud-top temperatures were generally higher for coupled clouds during warm-air events. In addition, the top temperature of precipitating liquid clouds do not fall below -10°C and don't show the tail towards low cloud-top temperatures found for non-precipitating liquid clouds.

Mixed-phase clouds during WAE were generally thinner than during non-WAE (Figures 10a and c). The deepest mixed-phase clouds were found for non-WAE and for decoupled PBLs. No difference is found in the thickness of mixed-phase clouds below the inversion for coupled and decoupled PBLs suggesting little difference in the geometrical properties of those clouds. The frequency distributions of cloud-top temperatures are very similar for all non-precipitating mixed-phase clouds observed for non-WAE cases independent of PBL coupling, with a broad peak in occurrence between 0°C and -20°C. This distribution is shifted to warmer temperatures during WAE, with the warmest cloud tops found for clouds that extends through the inversion (Figure 10b). The cloud-top temperature for precipitating mixed-phase clouds in Figure 10d show a peak at -5°C for coupled PBLs during both WAE and non-WAE conditions. Fewer cases of cloud-top temperatures below -10°C are found for precipitating mixed-phase clouds compared to non-precipitating mixed phase clouds. All cloud cases (non-precipitating and precipitating liquid and mixed-phase) show warmest cloud tops for coupled PBLs during WAE conditions. This is related to the fact that the top of those clouds extend into the warmer air aloft (Tjernström et al., 2015). Two detailed examples can be found in the Supplementary Material.

Figure 11 provides a profile view of the LWC of the liquid clouds considered in Figure 9. The scaled altitude ranges from the base of the clouds (zero) to the cloud top (unity). All profiles have been interpolated to intervals of 0.1 scaled altitude. Non-precipitating liquid clouds show maximum LWC between 0.03 and 0.20 $\mathrm{g\,m^{-3}}$ within the upper quarter of the cloud. The LWC is lowest for clouds observed during WAE. The LWC increases for precipitating liquid clouds with maximum values of 0.08 $\mathrm{g\,m^{-3}}$ during WAE and 0.20 to 0.40 $\mathrm{g\,m^{-3}}$ during non-WAE conditions.

The profiles of LWC and IWC of the mixed-phase clouds considered in Figure 10 are shown in Figure 12. The maxima in LWC are lower for mixed-phase clouds compared to liquid clouds with values close to zero for some precipitating mixed-phase clouds. Mixed-phase clouds during WAE generally show a lower maximum in LWC compared to those observed during non-WAE, particularly when they are non-precipitating. The profiles of IWC in mixed-phase clouds (Figure 12c and d) are distinctly different from those of LWC. They show a wide range in maximum values with lowest IWC close to $0\,\mathrm{g\,m^{-3}}$ for clouds below the inversion and highest values of 0.02 to $0.05\,\mathrm{g\,m^{-3}}$ for clouds above or extending through the inversion during non-WAE conditions. Note that these are also the geometrically thinnest and thickest clouds, respectively (Figure 10). The IWC profile generally peaks at a height below the peak in the LWC profile – usually in the centre of the cloud but sometimes closer to cloud base, likely due to increasing particle sublimation as the crystals fall.

During non-WAE, liquid clouds below the inversion (i.e. with cloud top at or below PBL top) showed no statistically significant difference in LWP (two-sample t-test, $p < 0.05$) for coupled and de-coupled PBLs, with mean values of $24\pm62\,\mathrm{g\,m^{-2}}$ (median of $6\,\mathrm{g\,m^{-2}}$) and $22\pm41\,\mathrm{g\,m^{-2}}$ (median of $8\,\mathrm{g\,m^{-2}}$), respectively (not shown). For clouds below the inversion in coupled PBLs, 90% of cases showed LWP below $50\,\mathrm{g\,m^{-2}}$ while this number slightly decreases to 88% for clouds below the inversion in decoupled PBLs. This behaviour is consistent with the observations reported in Sotiropoulou et al. (2016).

Mixed-phase clouds in the same situation (non-WAE, below inversion) showed LWP behaviour for coupled and de-coupled PBLs opposite to that of liquid clouds. We find a statistically significant difference (two-sample t-test, $p < 0.05$) with mean values of $33\pm57\,\mathrm{g\,m^{-2}}$ (median of $13\,\mathrm{g\,m^{-2}}$) and $52\pm63\,\mathrm{g\,m^{-2}}$ (median of $32\,\mathrm{g\,m^{-2}}$), for coupled and de-coupled PBLs, respectively (not shown). For clouds below the inversion in coupled PBLs, 76% of cases showed LWP below $50\,\mathrm{g\,m^{-2}}$ while this number decreased to 64% for clouds below the inversion in decoupled PBLs. Interestingly, mixed-phase clouds below the inversion in decoupled PBLs were slightly thinner than in coupled PBLs (Figure 10) while little difference was found in their respective profiles of IWC (Figure 12).

## 4 Discussion

Cloud observations in the Arctic are scarce. The available data sets discussed below are from different geographic regions, represent different time periods, and were obtained using different retrieval methods. Consequently, care must be taken when comparing them. Additional constraints apply when also considering spaceborne cloud observations. For instance, the CloudSat nominal blind zone of about 0.75 to 1.25 km from the surface (Tanelli et al., 2008) means that a large fraction of Arctic clouds cannot be accurately detected in CloudSat observations. Mech et al. (2019) analysed airborne microwave radar and radiometer measurements near Svalbard during ACLOUD (Wendisch et al., 2019) to find that about 40% of all clouds show cloud tops below 1000 m height, and thus, are likely to be missed by CloudSat. Nomokonova et al. (2019) find a peak frequency of cloud occurrence at 800 to 900 m from Cloudnet observations at Ny Alesund. In the case of ACSE, 50% and 37% of all clouds show cloud tops below 1000 m in summer and autumn, respectively. These numbers increase to 80% and 76% for liquid clouds. About 25% and 41% of mixed-phase clouds are affected during summer and winter, respectively. The effect is smallest for ice clouds with 5% during summer and 14% of observations during autumn.

Figure 13 compares the cloud-fraction profiles derived from the ACSE observations (left panel of Figure 5) to those reported for observations from ASCOS, conducted during August and early September 2008 well within the ice pack in the central Arctic Ocean. ASCOS cloud fractions were obtained following Shupe (2007). The profiles of total cloud fraction are very similar in shape but show a generally lower cloudiness from ACSE. Note that while the profiles represent roughly the same period of the year, the actual observations have been performed at different locations and in different years. Nevertheless, the

resemblance in the shape of the total cloud fraction profile indicates the usefulness of relating Arctic observations to each other; particularly given their scarcity. For the comparison of cloud fraction, we need to keep in mind that the upper measurement height during ACSE was restricted to 6 km by instrument settings. This constrains all cloud fractions to zero at and above 6 km, as we only consider clouds for which a cloud top has been detected below this height. The total cloud fraction for all clouds including those with undetected top heights, i.e. top heights above 6 km, is given by the grey dashed line for reference.

The cloud-fraction profile for liquid-only clouds during ACSE generally resembles the profiles derived from ASCOS measurements. However, the occurrence of liquid-only clouds was much lower during ACSE, except for the frequent fog periods in the lowermost height bins during the summer months. The occurrence of ice and mixed-phase clouds during ACSE also appear to be quite similar to those obtained from ASCOS. Considering that most of the clouds with undetected tops are likely to be ice clouds and that the shape of the cloud-fraction profile for mixed-phase clouds during ACSE resembles that of ASCOS,

Figure 13 shows that the height from which ice clouds are the dominant cloud type was about 1 km lower for ACSE than for ASCOS.

    The monthly total cloud fraction of 95% in July, 74% in August and 97% in September as observed during ACSE can also be put into the context of previous studies. Shupe (2011) compared observation from surface land sites (Figure 2) in Atqasuk (ceilometer, microwave radiometer), Barrow (ceilometer, radar, micro-pulse lidar, microwave radiometer, Atmospheric Emitted

Radiance Interferometer), Eureka (radar, high spectral resolution lidar, micro-pulse lidar, microwave radiometer, Atmospheric Emitted Radiance Interferometer), and the SHEBA project (ceilometer, radar, microwave radiometer, Atmospheric Emitted Radiance Interferometer). For July to September, they present a total cloud fraction of 92% to 98% at Barrow and Sheba. Lower values of 80% to 85% are given for Atqasuk, while increasing from 65% in July to 80% in August and September at Eureka. Zygmuntowska et al. (2012) and Mioche et al. (2015) used data from the Cloud Profiling Radar (CPR) aboard

the CloudSat satellite (Stephens et al., 2008) and the Cloud-Aerosol Lidar with Orthogonal Polarization (CALIOP) on the Cloud-Aerosol Lidar and Infrared Pathfinder Satellite Observations (CALIPSO, Winker et al. (2010)) satellite for the years 2007 and 2008, and the period from 2007 to 2010, respectively, to investigate total cloud fraction in the Arctic region. They find consistent values of 75% to 80% in July, 80% to 87% in August, and 84% to 90% in September. For all clouds, ACSE observations of more than 90% during July and September are mostly in line with the high cloud fractions observed during

SHEBA (Shupe, 2011).

    Cloud fractions of 60% to 90% as observed at Eureka (Shupe, 2011) and for the Arctic region (Zygmuntowska et al., 2012; Mioche et al., 2015) suggest that the ACSE finding of a total cloud fraction of 74% in August is well within the range of values one would expect for the Arctic region. However, it should be noted that spaceborne data sets provide better spatial coverage than ground-based measurements during ACSE, and thus, are more representative of average conditions. When comparing the

fraction of mixed-phase clouds observed during ACSE to the multi-year (2007 to 2010) CALIPSO/CloudSat data set analysed by Mioche et al. (2015) it is apparent that the ground-based ACSE observations during July with a mixed-phase cloud fraction of 51% are in general agreement with the data from spaceborne instruments. However, ACSE observations of 33% during August and 80% during September show significantly lower and higher, respectively, fractions of mixed-phase clouds than the satellite record. This is probably the result of natural variability combined with the effect of comparing local measurements during ACSE to area averaged results from satellite. Considering the fraction of mixed-phase clouds at Barrow, Eureka and SHEBA (Shupe, 2011), ACSE findings are in line with SHEBA values of around 50% during July and around 85% during September. However, the ACSE mixed-phase cloud fraction of 33% during August is much lower than the SHEBA observation of around 80% (see Figure 2 in Shupe (2011)). The lower August mixed-phase cloud fraction during ACSE does, however, resemble the findings for Barrow and Eureka (Shupe, 2011).

Figure 14 compares the connection between the fraction of ice-containing clouds and cloud-top temperature for clouds observed during ACSE with those reported by Zhang et al. (2010) and Bühl et al. (2013). These previous studies combine measurements with cloud radar and aerosol lidar from space and ground, respectively. As in this study, they analyse clouds on a profile-by-profile basis. However, Zhang et al. (2010) and Bühl et al. (2013) focused on mixed-phase clouds at mid-latitudes. While they find that about 50% of all clouds are mixed-phase at a temperature of about -10°C, the ACSE observations reveal that in the Arctic a mixed-phase cloud fraction of 50% is reached already at -2°C. Because Arctic clouds often occur in the form of multi-layered clouds (Liu et al., 2012b), it is most likely that ice-crystal seeding from upper-level clouds into lower-level clouds leads to the high mixed-phase cloud fraction at relatively high temperatures (Vassel et al., 2019). In addition, the statistics are influenced by clouds that extend through the inversion and into warmer air above during warm-air events. Figure 14 also shows that the clouds with the warmest cloud-top temperature are also precipitating. Previous studies suggest that almost all non-cirrus clouds with cloud top temperatures below -20°C are mixed-phase at mid-latitudes. In the Arctic, this is the case already for warmer cloud-top temperatures of -12°C.; though ice-containing cloud fractions for non-precipitating clouds with top temperatures below -18°C to -25°C were found to be lower than at mid-latitudes for ACSE observations during autumn.

Figure 15 puts the ACSE observations of LWP and IWP for clouds during summer and autumn into the context of the earlier observations of SHEBA and ASCOS. ACSE LWP frequency distributions – though different for summer and autumn – do not resemble the previous observations, having a wider distribution with less well defined peak. The ACSE observations of IWP closely follow the ASCOS frequency distribution, although with larger values in the tail. There was quite a substantial part of the ASCOS ice drift during which mixed-phase stratocumulus clouds dominated, that may bias ASCOS LWP statistics high. In addition, air mass transit time is known to be an important factor in boundary layer structure and hence cloud properties. The fact that SHEBA and ASCOS have been further away from open water than ACSE means that air mass transit time is a factor controlling the cloud properties observed.

A comparison of ACSE remote-sensing observations to airborne in-situ measurements available in the literature is not straightforward due to differences in location and covered time period. In addition, it is challenging to ensure that clouds probed during in-situ observations are comparable to the clouds probed with remote-sensing observations. Nevertheless, we

have added a comparison to the in-situ profiles of cloud temperature, LWC, and IWC for single-layer mixed-phase clouds presented in Mioche et al. (2017) to the Supplementary Material in which we find similar shapes of the LWC and IWC profiles.

## 5    Summary and Conclusions

We present remote-sensing observations of Arctic clouds conducted during a three-month cruise in the Arctic Ocean along the Russian shelf from Tromsø, Norway, to Barrow, Alaska, and back. Observations with ceilometer, Doppler lidar, cloud radar
and microwave radiometer were made within pack ice, open water, and the marginal ice zone. The Cloudnet retrieval has been applied to investigate cloud properties with special emphasis on the effects of cloud-top temperature and boundary layer structure. The data set has been split into summer and autumn based on a change in the lower tropospheric mean temperature observed from radiosoundings launched every 6 hours (Sotiropoulou et al., 2016).

The ACSE data set reveals a strong reduction in the occurrence rate of liquid clouds and an increase for both mixed-phase
clouds and ice clouds at low levels during autumn compared to summer. Ice clouds during autumn extend almost down to the surface, while low clouds during summer are predominantly liquid. In addition, it was found that liquid clouds vary little in their vertical extent between summer and autumn, while both ice and mixed-phase clouds have lower top and base heights in autumn than in summer.

About 74% of all liquid clouds observed during ACSE show LWP below the infra-red black body limit of approximately
440 $50\,\mathrm{g\,m^{-2}}$. This means that the majority of the observed Arctic liquid water clouds have long-wave radiative properties that are highly sensitive to small changes in LWP. In general, the frequency distribution of LWP shows little variation for mixed-phase and purely liquid clouds. Nevertheless, summer shows more cases of high LWP and fewer cases of low LWP and the mean and median values are higher for mixed-phase clouds. The majority of clouds had an IWP below $20\,\mathrm{g\,m^{-2}}$ with summer (autumn) mean and median values of 34 and $7\,\mathrm{g\,m^{-2}}$ (32 and $9\,\mathrm{g\,m^{-2}}$), respectively.

Whether the PBL structure was coupled or decoupled, and the occurrence of warm air advection had little effect on the geometric thickness of liquid clouds. In contrast, mixed-phase clouds during WAE are generally thinner than for non-WAE. The deepest mixed-phase clouds are found for non-WAE and for decoupled PBLs.

Cloud-top temperatures for all mixed-phase clouds during non-WAE are between 0°C and -30°C. This range is reduced to 0°C to -20°C for mixed-phase clouds during WAE.

For liquid water clouds, the normalised profile of LWC reveals a strong increase with height with a maximum between 0.03 and $0.19\,\mathrm{g\,m^{-3}}$ within the upper quarter of the clouds followed by a sharp decrease towards cloud top. LWC is lowest for clouds observed below the inversion during WAE. Most mixed-phase clouds show a LWC profile with a very similar shape to that of liquid clouds with lower maximum values during WAE than during non-WAE.

The normalised profiles of IWC in mixed-phase clouds look different from that of LWC. They show a wider range in
maximum values with lowest IWC for clouds below the inversion and highest values for clouds above or extending through the inversion. Note that these correspond to the thinnest and thickest clouds, respectively. The IWC profile generally peaks at

a height below the peak in the LWC profile – usually in the centre of the cloud but also closer to cloud base and likely due to more particle sublimation as the crystals fall.

Unsurprisingly, it was found that liquid-water clouds during summer show the highest cloud-top temperatures, which can exceed 15°C but don't go below -15°C. As documented in Tjernström et al. (2015, 2019), ACSE cloud-top temperatures above 10°C correspond to a period of strong warm air advection that occurred at the beginning of August 2015. As a consequence, the frequency distribution of cloud-top temperature observed during summer resembles that derived from Cloudnet observations at mid-latitudes (Bühl et al., 2016). In autumn the top temperatures of liquid clouds rarely exceed 0°C with observed values as low as -25°C. The maximum of cloud-top-temperature occurrence rate shifts from 0°C in summer to -5.0°C in autumn.

During summer, the tops of mixed-phase clouds are generally warmer than in autumn with a maximum just below 0°C. However, they are always colder than liquid-only clouds during the same season. During summer, cloud-top temperature can be as low as -25°C though they are mostly warmer than -10°C. Autumn reveals a bi-modal distribution of cloud-top temperature corresponding to precipitating ($T_{top}$ >-10°C) versus non-precipitating clouds ($T_{top}$ <-10°C).

The IWC of mixed-phase clouds during summer and autumn mostly feature very low IWC of less than $0.07\,\mathrm{g\,m^{-3}}$, though values exceeding $100\,\mathrm{g\,m^{-3}}$ have been observed during autumn. In general, IWC was lowest in clouds with a low cloud top height and highest for clouds with top heights in the range from 3.0 to 4.0 km.

While the three-month ACSE data set provides comprehensive observations of Arctic clouds, it is challenging to relate the findings to earlier campaigns such as SHEBA or ASCOS. Although we find similar frequency distributions of LWP and IWP, the occurrence rate of clouds during ACSE was lower than during ASCOS. On the one hand, the observations have been conducted in different regions of the Arctic; consequently, observed differences might be the result of regional effects. On the other hand, different campaigns cover different time periods. This means that inter-annual variability might be added on top of potential regional effects – this is particularly highlighted by the warm air advection events observed during ACSE.

*Data availability.* Data from ACSE are available through the Bolin Centre for Climate Research (http://www.bolin.su.se), the Centre for Environmental Data Analysis (https://www.ceda.ac.uk/), and the National Oceanic and Atmospheric Administration (ftp://ftp1.esrl.noaa.gov/psd3/cruises/SW

*Author contributions.* PA has analysed the ACSE data set and prepared the manuscript together with IMB and MT. PA has set up the Cloudnet retrieval at Leeds together with EJO. GS has provided the inversion heights from radiosounding. IMB, BJB, GS, MDS, and MT performed the measurements during ACSE. BP has assisted in refining the MWR retrieval for Arctic observations. All authors contributed to the discussion of the results and revision of the manuscript.

*Competing interests.* The authors declare no competing interests.

*Acknowledgements.* ACSE was supported by funding from the Knut and Alice Wallenberg Foundation, Swedish Research Council, Faculty of Science at Stockholm University, US Office of Naval Research, the US National Oceanic and Atmospheric Administration (NOAA), and the UK Natural Environment Research Council (grant No. NE/K011820/1). We are grateful to the Swedish Polar Research Secretariat and to the two captains and crews of the Oden for logistics support. The radiosounding system, Halo lidar, and HATPRO radiometer were supplied by the National Centre for Atmospheric Science (NCAS) Atmospheric Measurement Facility. The cloud radar was provided by NOAA. PA would like to thank Richard Rigby for Linux support while setting up the Cloudnet retrieval.

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

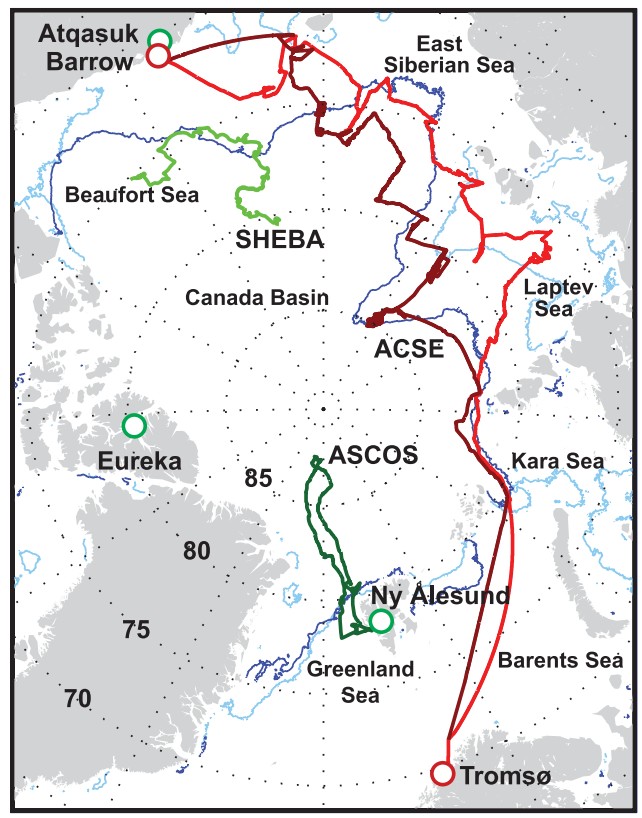

**Figure 1.** Map of the ACSE cruise track (leg 1 in red, leg 2 in burgundy) together with the sea ice extent on 5 July 2014 (light blue) and 5 October 2014 (dark blue). The tracks of the ASCOS and SHEBA experiments are given in dark and light green, respectively. Red circles mark the start and end of the ACSE cruise track. Green circles give the location of other Arctic sites referred to in this paper.

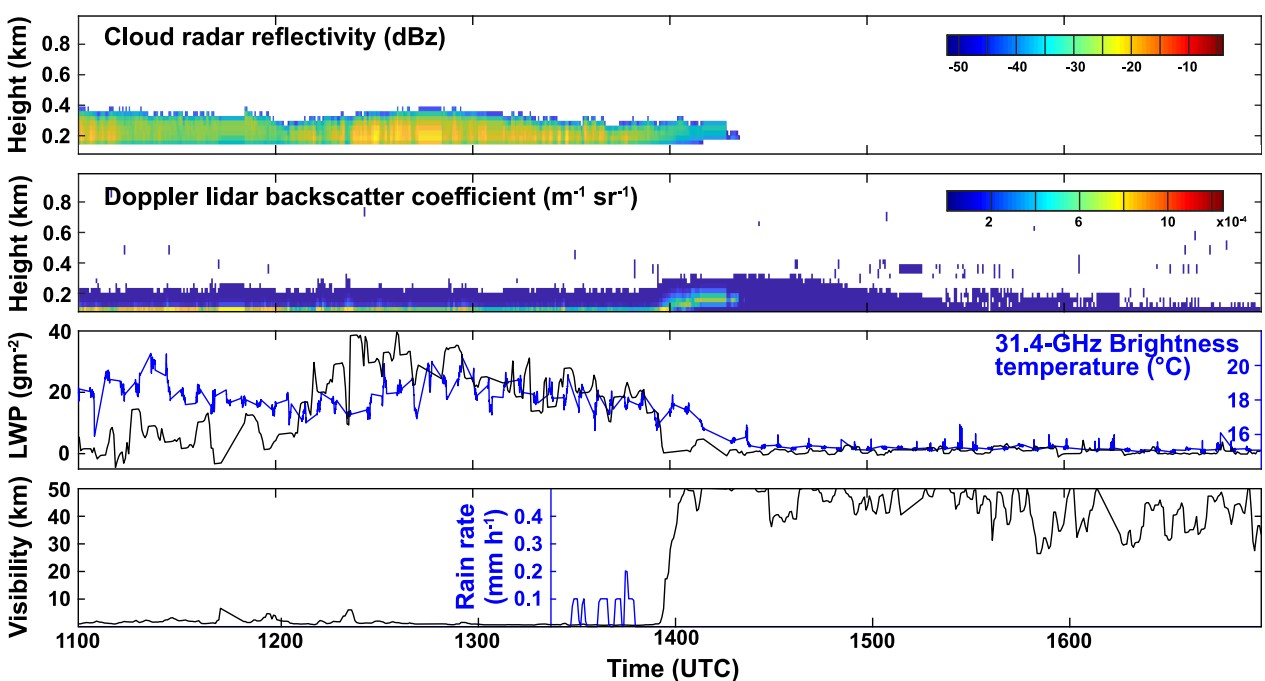

**Figure 2.** Time series of cloud radar reflectivity, Doppler lidar backscatter coefficient, LWP, 31.4-GHz brightness temperature, visibility, and rain rate for the time period from 1100 to 1700 UTC on 25 July 2014.

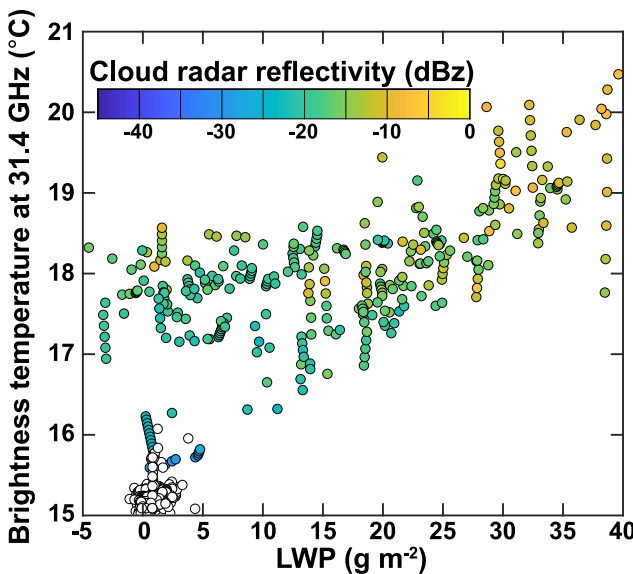

**Figure 3.** Scatter plot of the change in brightness temperature at 31.4 GHz as measured by the MWR with LWP. Colour coding refers to the maximum in cloud radar reflectivity in dBz for the respective profile. The data cover the time period from 1100 to 1700 UTC on 25 July 2014 as in Figure 1. White circles refer to cloud-free regions.

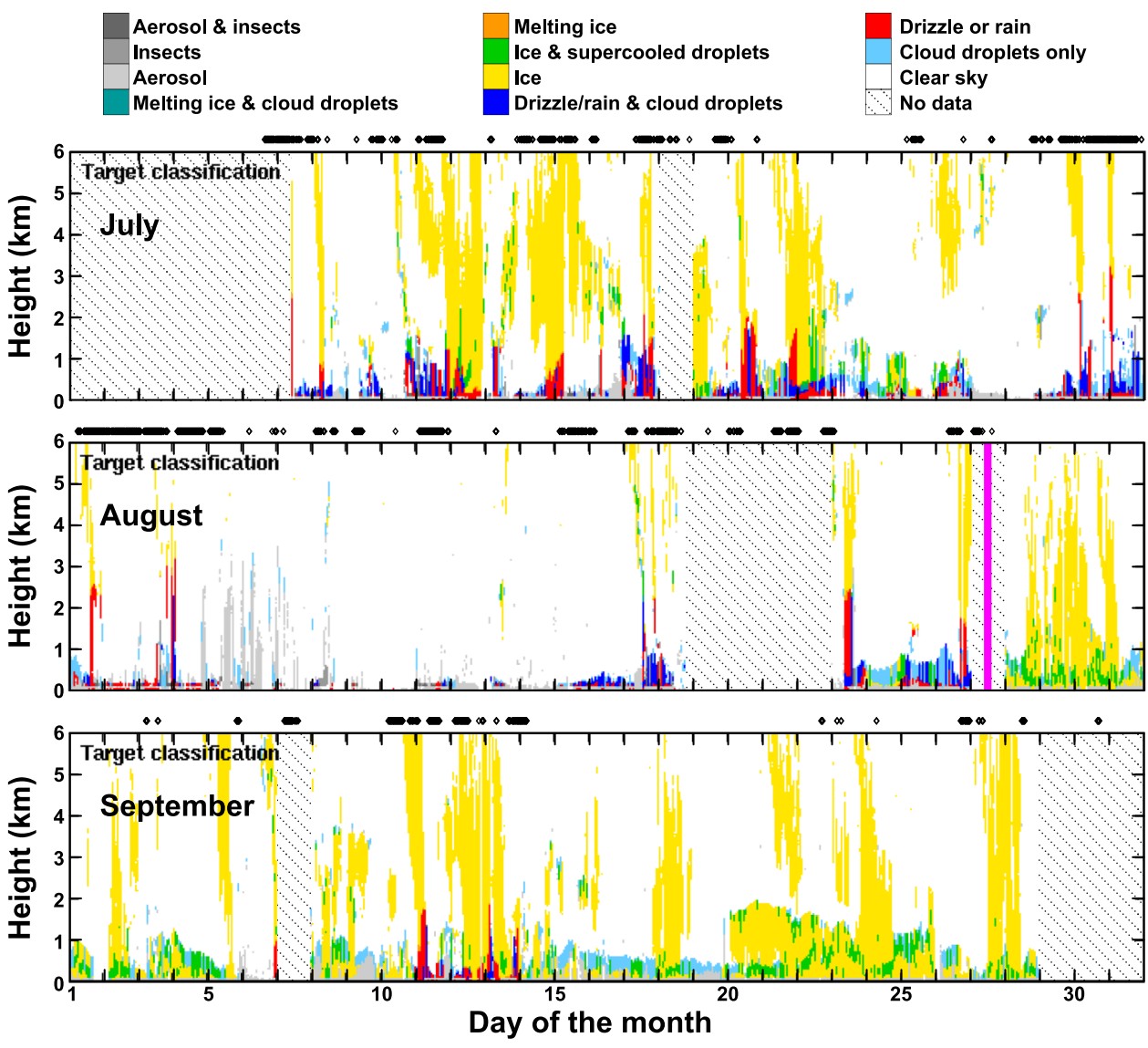

**Figure 4.** Cloudnet target classification for the three-month ACSE cruise. Black diamonds above the monthly displays mark 10-min periods of visibility below 1 km. Hatched areas separate periods of no data from the white background of *Clear sky*. The pink vertical line on 27 August 2014 marks the transition from summer to autumn as defined by Sotiropoulou et al. (2016).

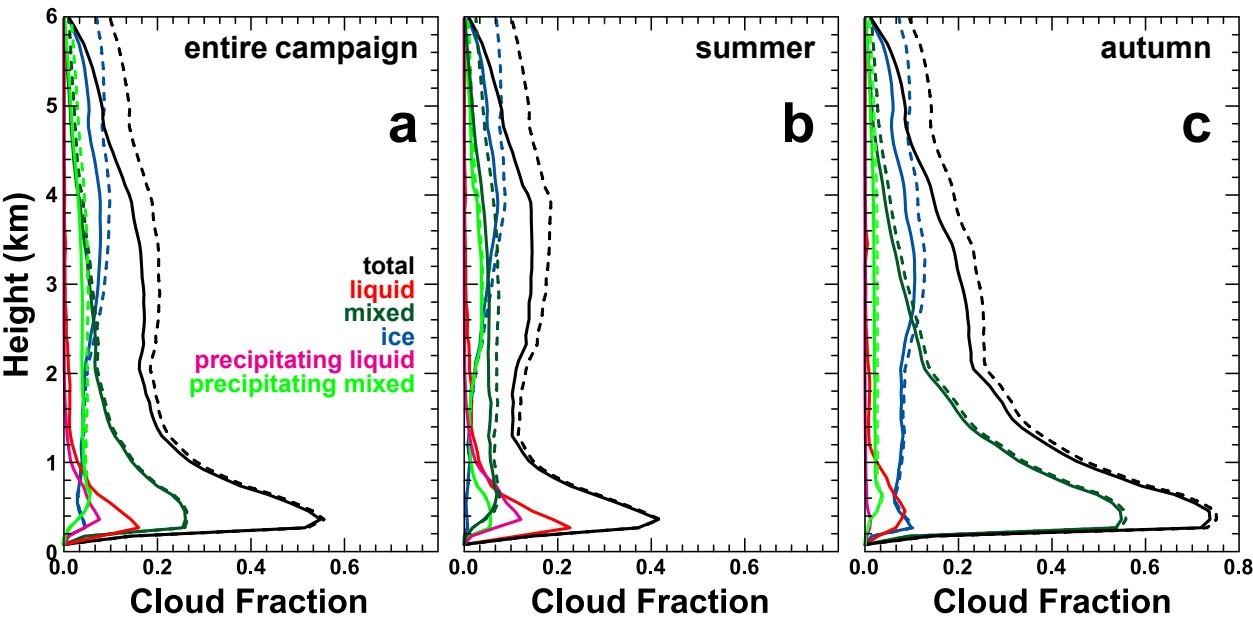

**Figure 5.** Profiles of cloud fraction for different cloud types as obtained using Cloudnet for the entire ACSE campaign (left), summer (middle), and autumn (right). All solid profiles refer to clouds for which the cloud-top height was located below 6 km height. The dashed lines refer to the total cloud fraction with respect to all clouds, i.e. including those with undetected cloud-top heights.

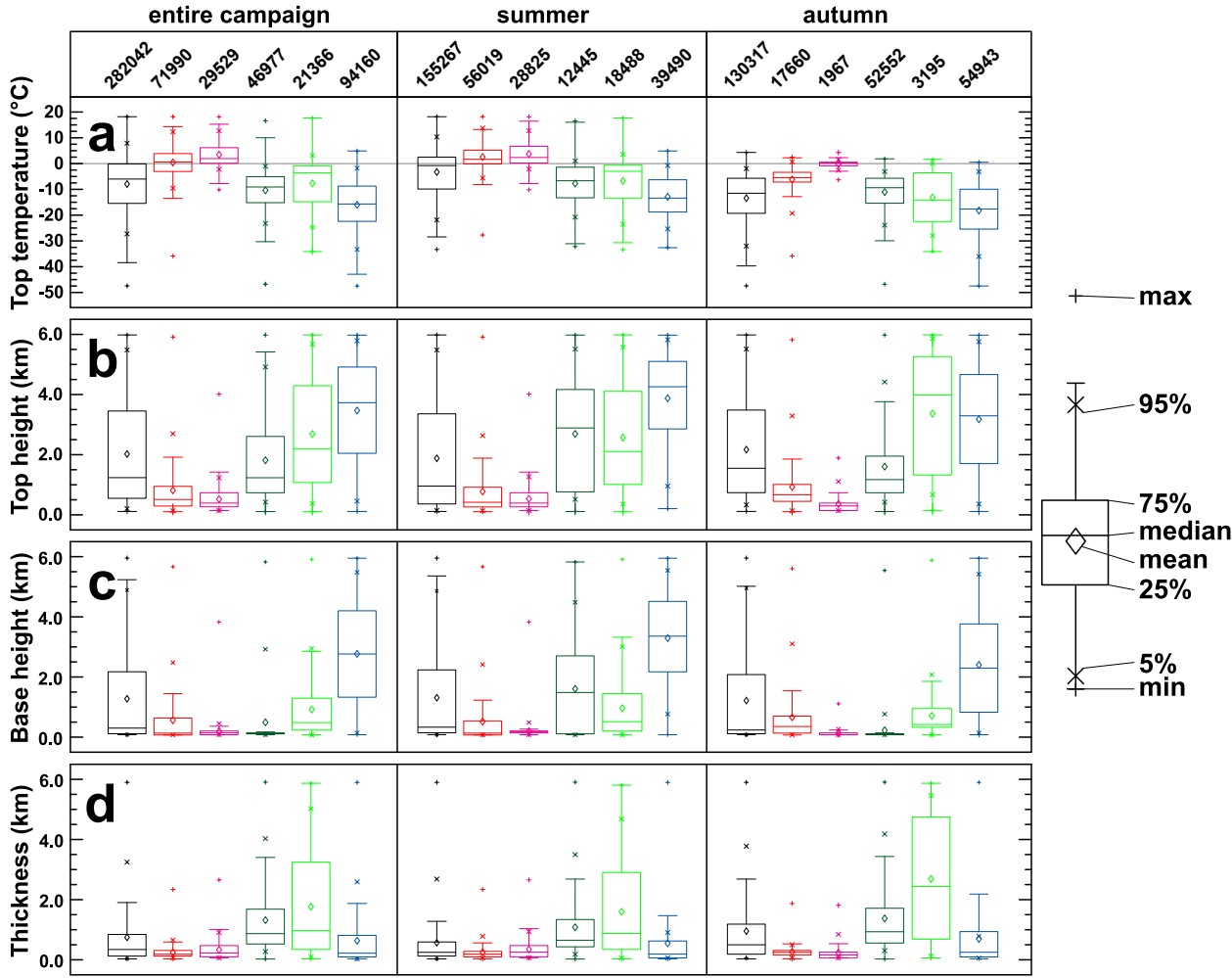

**Figure 6.** Statistical overview of cloud occurrence with respect to (a) top temperature, (b) top height, (c) base height, and (d) geometrical thickness for the entire ACSE campaign (left column) as well as for summer (middle column) and autumn (right column). The colours indicate the different cloud types as in Figure 3: all (black), non-precipitating liquid (red), precipitating liquid (magenta), non-precipitating mixed-phase (dark green), precipitating mixed-phase (light green), and ice (blue). The numbers in the top panel refer to the number of 30-s profiles considered in the analysis.

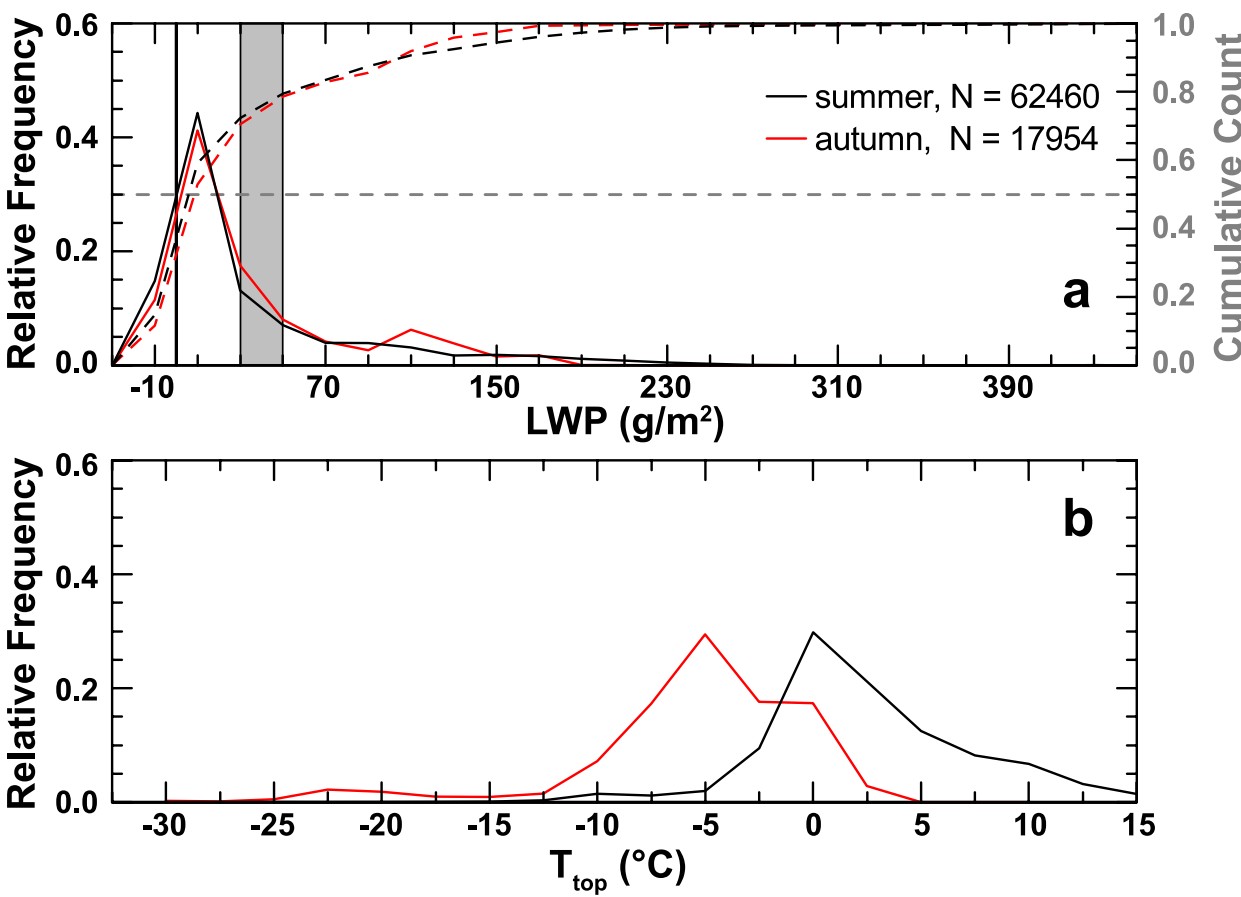

**Figure 7.** (a) Histogram (solid lines) and cumulative count (dashed lines) of the occurrence frequency of liquid water path, and (b) histogram of the cloud top temperature for liquid clouds observed during summer (black) and autumn (red). Values represent individual cloud layers on a profile basis. The grey dashed line in (a) marks 50% in the cumulative counts. The vertical line in (a) marks $0\,\mathrm{g\,m}^{-2}$ LWP while the grey area indicates the infrared black body limit between 30 and $50\,\mathrm{g\,m}^{-2}$ LWP.

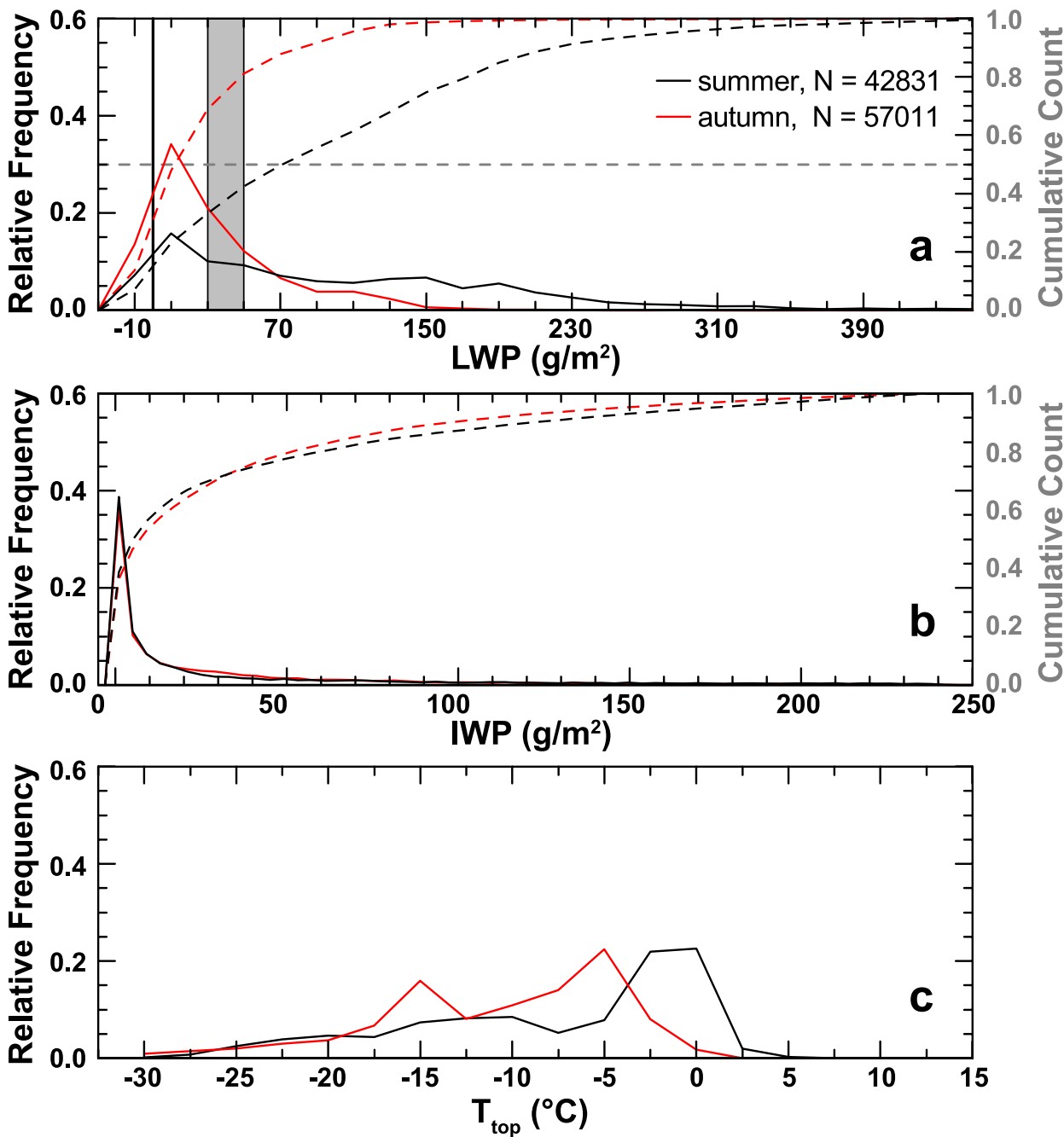

**Figure 8.** Histogram (solid lines) and cumulative count (dashed lines) of the occurrence frequency of liquid water path, ice water path as well as the histogram of the cloud top temperature for mixed-phase clouds observed during summer (black) and autumn (red). Values give the number of considered cloud layers as observed on a profile basis. The grey dashed line in (a) marks 50% in the cumulative counts. The vertical lines in (a) mark 0 and $50\,\mathrm{g m^{-2}}$ LWP. The latter is the infrared black body limit.

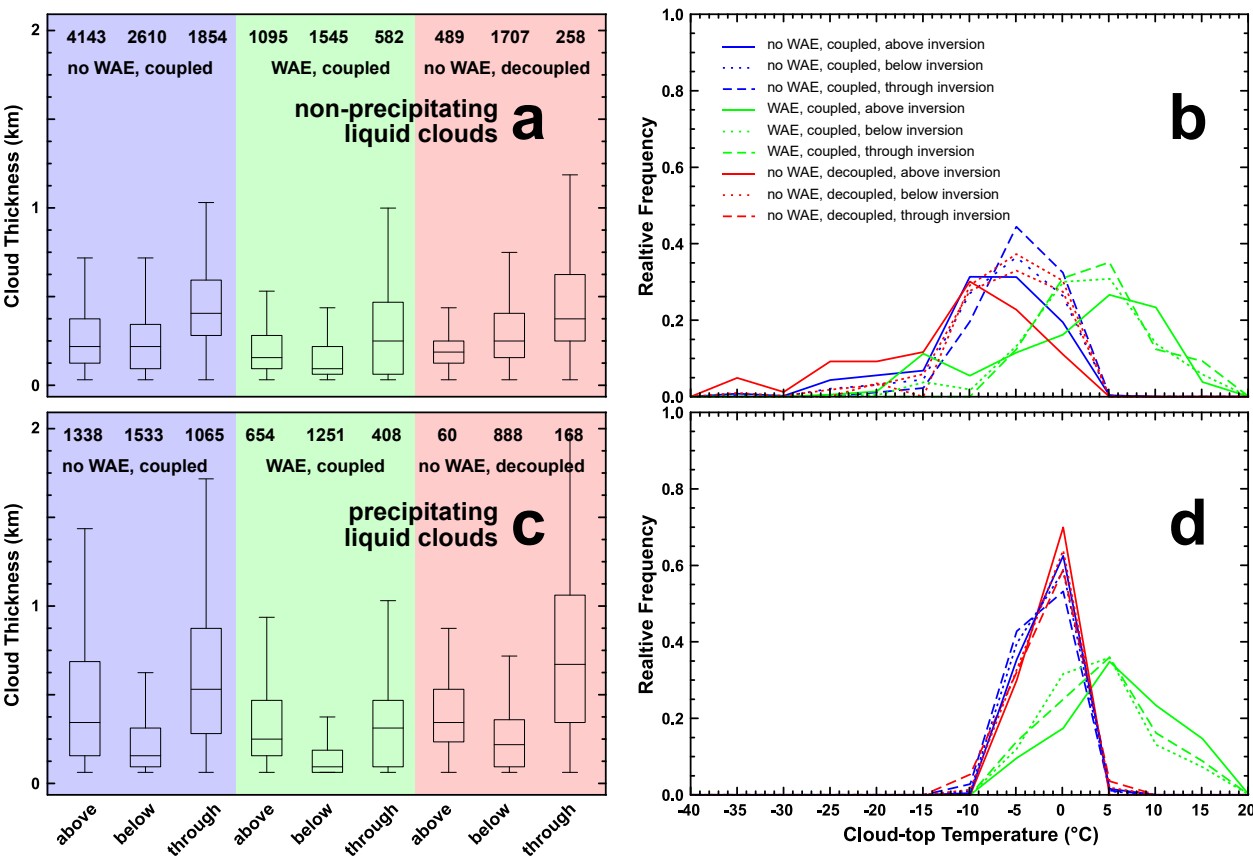

**Figure 9.** Statistics on the geometrical thickness (a and c) and the frequency distribution of cloud-top-temperature (b and d) of non-precipitating (a and b) and precipitating liquid clouds (c and d) clouds observed for different PBL structure, temperature in the free troposphere, and location with respect to the main inversion: non-WAE with coupled PBL (blue), non-WAE with decoupled PBL (red), and WAE with coupled PBL (green). The different boxes (a and b) and lines in (c and d) refer to clouds with cloud base above the inversion (above), to clouds with cloud top below the inversion (below) or to clouds with cloud base below the inversion and cloud top above the inversion (through). Numbers in (a) and (b) refer to the number of Cloudnet profiles per category. Categories with less than 100 profiles have been omitted; this includes all cases of decoupled PBL during WAE.

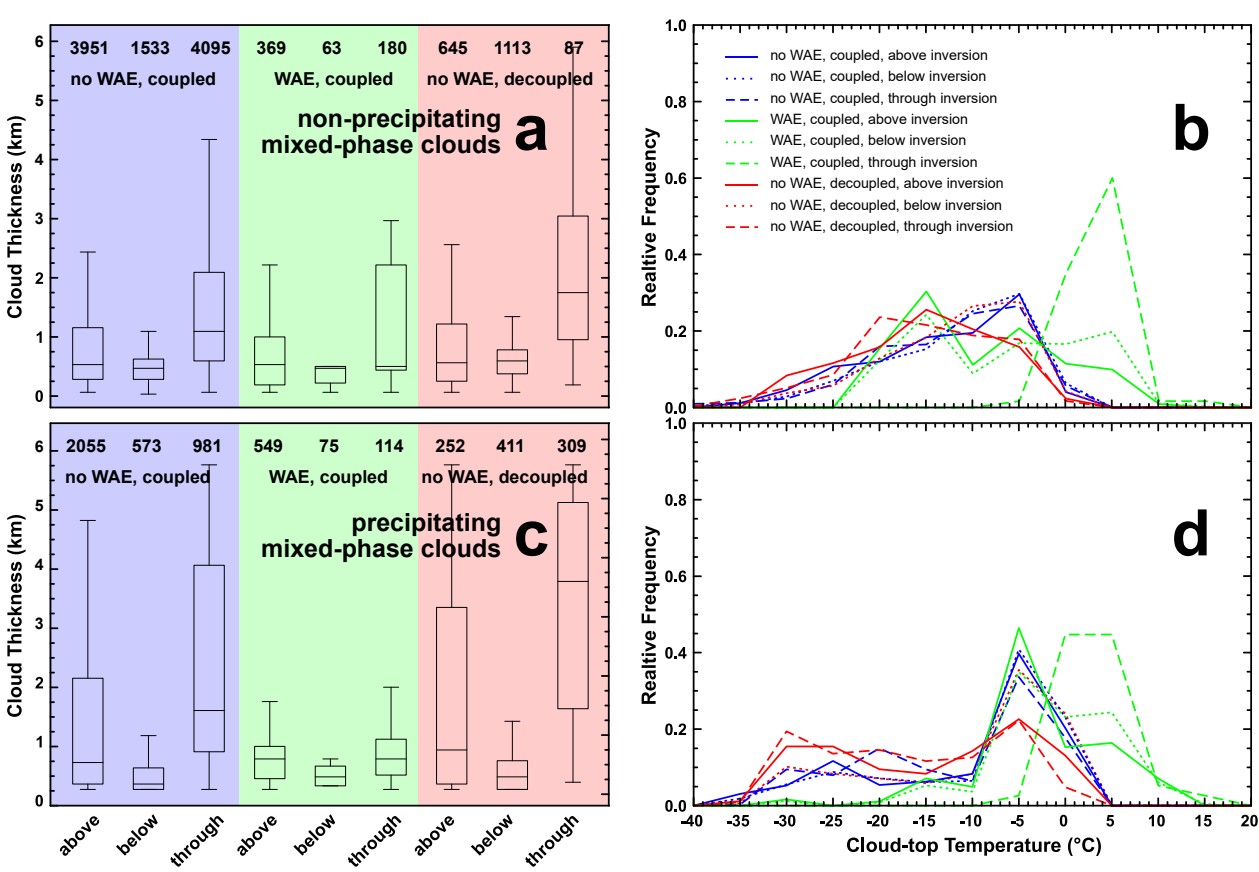

**Figure 10.** Same as Figure 9 but for non-precipitating (a and b) and precipitating mixed-phase clouds (c and d).

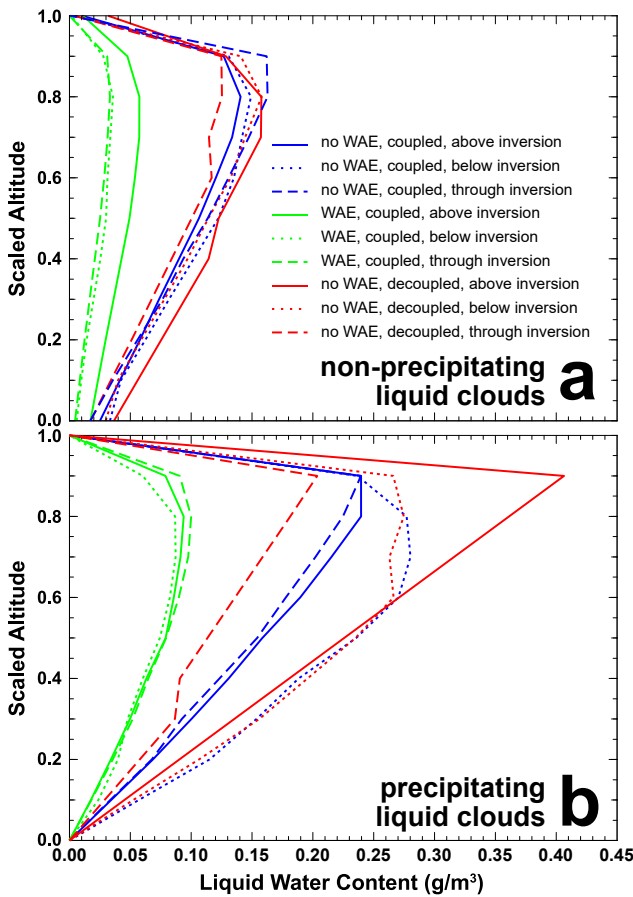

**Figure 11.** Scaled profiles of LWC for non-precipitating (a) and precipitating liquid clouds (b) observed for different PBL structure, temperature at 1 km height, and location with respect to the main inversion. Zero and unity of the scaled altitude refer to cloud base and top, respectively. Categories with less than 100 profiles have not been included.

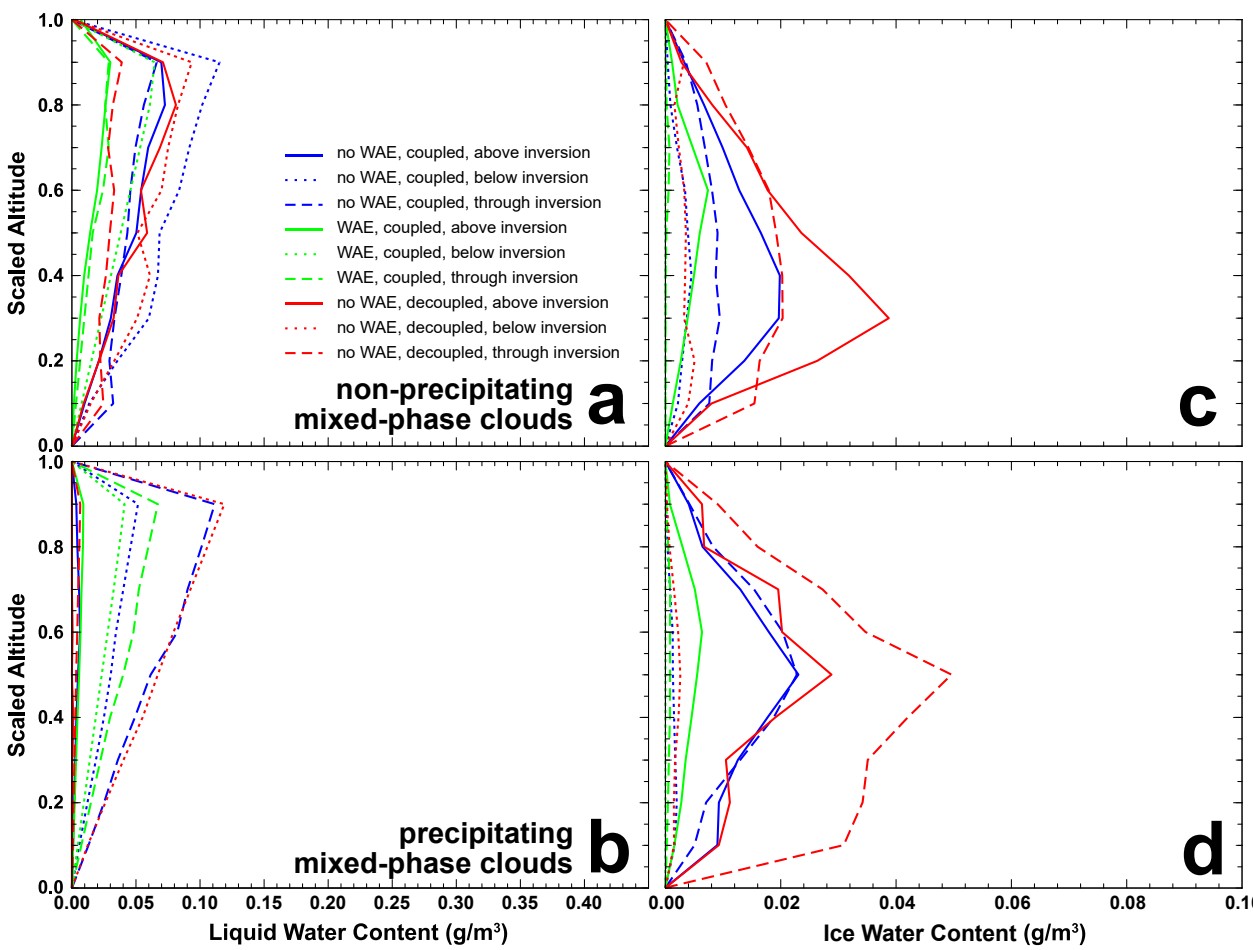

**Figure 12.** Scaled profiles of LWC (a and b) and IWC (c and d) for non-precipitating (a and c) and precipitating mixed-phase clouds (b and d) observed for different PBL structure, temperature at 1 km height, and location with respect to the main inversion. Zero and unity of the scaled altitude refer to cloud base and top, respectively. Categories with less than 100 profiles have not been included.

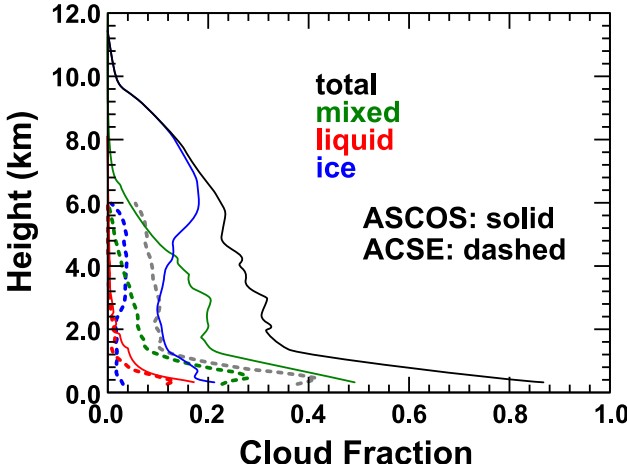

**Figure 13.** Profiles of cloud fraction for different cloud types as derived from measurements during ASCOS (solid, 12 August to 2 September 2008) and ACSE (dashed) for the ASCOS time period. The grey dashed lines refers to all clouds, i.e. including those for which cloud top extended above the maximum measurement range of 6 km height.

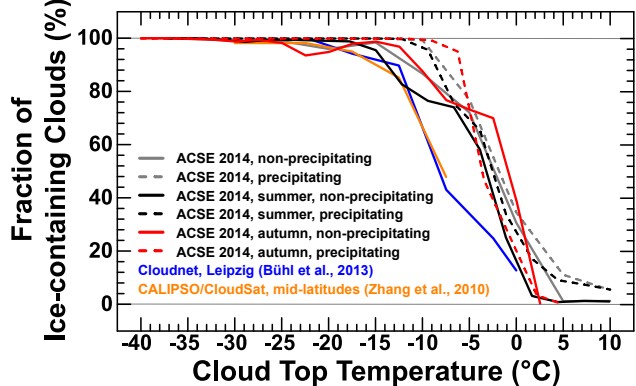

**Figure 14.** Fraction of non-precipitating (solid) and precipitating (dashed) mixed-phase clouds observed during ACSE in summer (black) and autumn (red) in comparison to the entire ACSE data set (gray) as well as to previous observations of mid-level clouds at mid-latitudes from ground (Bühl et al., 2013) and space (Zhang et al., 2010).

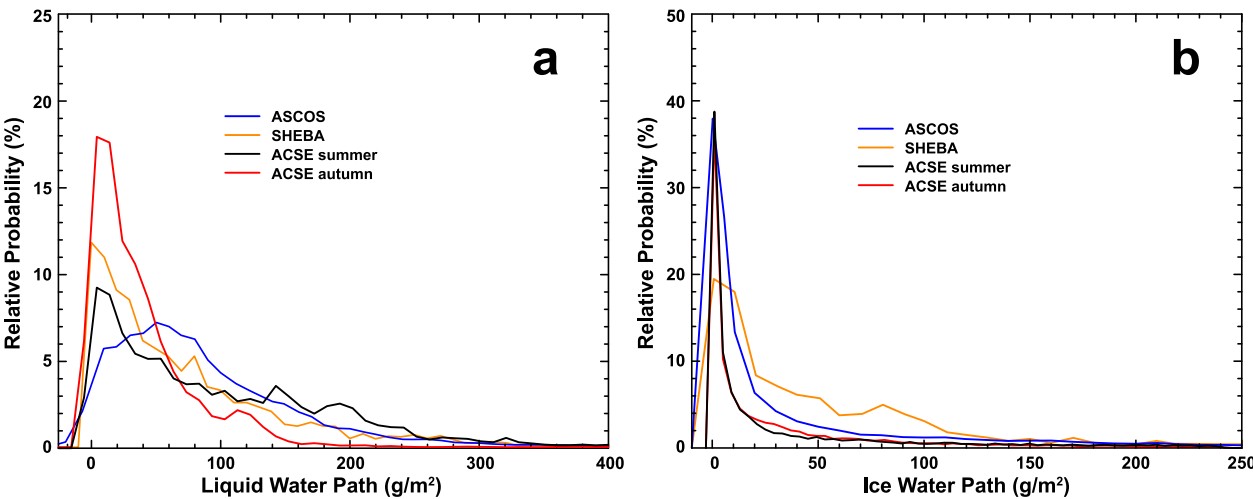

**Figure 15.** Relative probability of (a) LWP and (b) IWP for mixed-phase clouds observed during ACSE in summer (black) and autumn (red) in comparison to previous observations during ASCOS (blue) and SHEBA (orange). Figure adapted from Figure 17 in Tjernström et al. (2012).