# Peer review of "Properties of Arctic liquid and mixed phase clouds from ship-borne Cloudnet observations during ACSE 2014"

_Atmospheric Chemistry and Physics, 2020_

## Referee Comment (RC1) · Anonymous Referee #1 · 16 Feb 2020

This is a descriptive paper that examines data collected during a Summer and Autumn cruise along the Siberian Shelf in 2014. The purpose of the study is to document the statistics of the clouds that were observed with a focus on their geometric properties and bulk microphysical characteristics. There was emphasis on how those properties changed from Summer to Autumn. It is difficult to find a unifying theme to this paper aside from the fact that clouds change between summer and fall. Some of my difficulty stems from an insufficient description of the measurements and the algorithms and why the authors chose to focus on some data streams but not others.

This paper would be much helped by demonstrating a couple of illustrative case studies

that show typical events and that illuminate the main points of the paper. For instance, I would like to know what the change in microwave brightness temperature at 31 GHz looks like for a cloud with liquid water path of 10 g m-2, how such a cloud appears on the w-band radar and whether the lidar is penetrating through the cloud. Another useful case study would illustrate a cloud system that extends through the inversion and that is precipitating ice versus liquid.

Precipitation is not addressed at all yet the boundary layer clouds at issue here must precipitate quite often.

1. While the occurrence of semitransparent liquid clouds in the Arctic have been known for some time, these are normally diagnosed with an infrared spectrometer and ceilometer. Is there something special about the microwave radiometer used in this study and the algorithm used in the inversions that allow it to have precision below the 20 or 30 g m-2 that is typical of such instruments? This seems to be the level of precision cited in the referenced papers. The authors describe an "offset correction" for the LWP based on clear sky periods. What does this mean and how is determined and justified?

2. The authors use an algorithm to categorize profiles that was published in 2007 as part of the Cloudnet program. I am skeptical that this algorithm could be used reliably without modification from one data set to another since such algorithms always rely on various tunable thresholds to make decisions. I wonder if the authors could describe the thresholds used by this algorithm to classify profiles between liquid, mixed, and ice and whether they found it necessary to adjust the thresholds in the algorithm.

3. The Hogan et al (2006) IWC algorithm uses airborne data collected over the UK in mostly frontal clouds as I recall using a Mass-Size relationship developed by Brown and Francis in such clouds. Is there a reason to suspect that the ice precipitation produced in the Arctic boundary layer is similar to frontal ice precipitation over the UK so that the same regression algorithm could be used here?

4. Perhaps I missed it but how often were radiosondes launched during this voyage?

5. A binary classification of warm air advection based on the temperature at 1 km seems problematic to me. Warm air advection implies large scale transport with the wind perpendicular to a thermal gradient. True warm air advection would have real implications for clouds because it is typically accompanied by deep large-scale ascent, etc. Perhaps the authors could show statistics of the strength of the advection in units of temperature per unit time and how the cloud distributions respond to this?

6. The vertical structure of mixed phase clouds is important. In these Arctic clouds typically ice precipitation sediments from a thin liquid water layer. Is that what is seen here? Is the radar able to detect the water layers? If so, how often? Is the MWR capable of retrieving the water path from such layers? If so, what is the typical increase in microwave brightness temperature? A case study would be interesting and useful to establish confidence in the analysis methodology.

7. There is no mention of drizzle, light rain or snow. I'm not sure how a study of clouds in the Arctic could be complete without considering the properties of the precipitation that certainly occurs much of the time.
* * *

---

## Referee Comment (RC2) · Anonymous Referee #2 · 1 Mar 2020

This is a useful manuscript that documents observations of liquid and ice in Arctic clouds that were made during the ACSE 2014 field experiment. Liquid and ice properties were retrieved from radar, radiosonde, and other observations using the Cloudnet algorithm.

There were no in-situ observations of these clouds, and so further discussion of measurement uncertainties or else comparison to in-situ aircraft observations of similar clouds in the Arctic or mid latitudes would be welcome.

The authors also might consider adding more speculations about the physical mechanisms behind the observations. For instance, why are the clouds mixed-phase at such

warm temperatures (Fig. 10)? More physical explanation might help better tie together the observations into a more coherent understanding.

Specific comments:

Lines 116–118: "Liquid water content (LWC) is calculated from microwave radiometer-derived LWP (with an offset correction based on clear-sky periods) and liquid layer cloud boundaries by distributing the liquid using the scaled-linear adiabatic assumption, i.e. LWC increasing linearly with height from zero at cloud base." So LWC is assumed to increase linearly with height, but then in Fig. 8, the LWC is plotted and the authors conclude "Liquid clouds show maximum LWC between 0.03 and 2.00 gm$-3$ within the upper quarter of the cloud". Is this a conclusion about nature or merely an artifact of the authors' assumption of linearly increasing LWC with height?

Lines 143–146: "We further separate the large-scale circulation between warm-air advection events (WAA, Tjernström et al. (2015) and conditions during which no warm-air advection took place (non-WAA). WAA was identified from the ACSE soundings as when the temperature at 1.0 km height exceeded a threshold of 5 C, empirically derived from Figure 2a of Sotiropoulou et al.(2016)." Is this simple temperature criterion sufficient to accurately diagnose warm-air advection? Can it be corroborated by, e.g., wind observations showing wind from southern, warmer regions?

Lines 247–248: "Figure 8 provides a profile view of the LWC and the IWC of the clouds considered in Figure 7. The scaled altitude ranges from the base of the clouds (zero) to the cloud top (unity)." In this figure, is the cloud base defined as liquid cloud base, ice cloud base, or the lower of the two? In the leftmost (liquid) panel, some clouds exhibit non-zero liquid at cloud base, which seems contradictory. Shouldn't cloud base in a liquid-only cloud be defined as the altitude at which liquid goes to zero? If so, how can the liquid remain non-zero at cloud base? In the middle and right-hand (mixed-phase) panels, all but two of the cloud layers have liquid extending all the way to cloud base, and ice approaching zero at cloud base. Why weren't there more clouds with ice falling

out of liquid cloud base? Is this an observational artifact?

Line 329–330: "While they find that about 50% of all clouds are mixed-phase at a temperature of about -10 C, the ACSE observations reveal that in the Arctic a mixed-phase cloud fraction of 50% is reached already at -2 C." Can the authors speculate about why this difference is observed in these datasets for Arctic and mid-latitude clouds? E.g., do more of the Arctic clouds have cold regions above that nucleate ice, which then sediments down to warmer temperatures (e.g., Fleishauer et al. 2002)?

Lines 350–351: "The ACSE data set reveals a strong reduction in the occurrence rate of liquid clouds and an increase for both mixed-phase clouds and ice clouds at low levels during autumn compared tosummer." To me, a striking difference between summer and autumn is that there is more drizzle/rain during summer (see Fig. 2). I encourage the authors to comment on this if they have a hypothesis why this is observed.

Fig. 1: Is it possible to display more clearly the area covered by sea ice using, e.g., light-blue shading?

Fig. 2: The red-green-blue (liquid-mixed-ice) colour convention of Figures 3 and 4 is not followed in Fig. 2, which uses yellow to denote ice, e.g.. Would it be clearer to make the colour convention of Fig. 2 match that of Figures 3 and 4?

Fig. 2 seems to suggest that ice clouds are geometrically thicker than mixed-phase clouds, whereas Fig. 4 seems to suggest that it's the other way around. Which is it? Are these two figures consistent with each other?

Reference:

Fleishauer, R. P., Larson, V. E., & Vonder Haar, T. H. (2002). Observed microphysical structure of midlevel, mixed-phase clouds. Journal of the atmospheric sciences, 59(11), 1779-1804.

---

## Referee Comment (RC3) · Anonymous Referee #3 · 23 Apr 2020

General global comment:

This paper documents the geometrical and microphysical properties of Arctic clouds from observations based on remote sensing instruments onboard the Oden vessel during the 3 months of the ACSE experiment. The authors focused on cloud properties during summer and autumn seasons and differences are highlighted.

I think the data and results presented in this manuscript are useful since Arctic clouds are still less documented then their mid-latitude or tropical counterparts. Moreover, the study area is very large since it extends from Tromso to Barrow, so the present results are representative of the Arctic region.

[Figure]

My main comments are below:

- My first main remark is that the authors could greatly enhance the discussion of the results by including the environmental conditions when comparing the cloud properties (and not limit only to the comparison between summer and autumn). Indeed, I think the paper would be greatly improved by a section presenting the meteorological conditions during ACSE. Not a detailed characterization of course, but at least some statistics about environmental parameters (temperature, humidity, winds, air mass origins) in order to identify the main regimes encountered during the experiment. In this new section, you may include for example the description of the WAA events.

- Moreover, since all the results are based on remote sensing measurements, I think the authors could develop the uncertainties and limitations of such instrumentation in more details.

- You compare your results with satellite observations, but not to in situ measurements. I would suggest to do it, keeping in mind that in situ measurements remain very localized in time and space.

- Line 96: Please, could you explain how is used the training dataset in the radiometer retrieval algorithm and why you could not use the radiosondes from ACSE? Moreover, in this section 2.2, you mention the radiosondes and the vertical profiles of temperature and humidity made during ACSE. I suggest the authors to show some statistics/figures about these data to present the context of your results (you may include them in the "meteorological conditions", cf. previous comment).

- The presented statistics are made for each identified cloud layer. Since you analyze up to three by profiles (cf. line 166), I think it could be interesting to calculate the ratio of single layer cloud vs multilayer cloud (with keeping in mind this maximum limit of 3 layers).

- In lines 122 to 131 you present the target classification. The readability would be

enhanced with a schemed summarizing this classification.

- Was there an instrument on board to measure the precipitation? It could be of great interest to identify the precipitating clouds from the non-precipitating. Precipitation is closely linked to cloud properties and is a key factor but still difficult to assess. Also, it would help the discussion, for example in lines 373-374 on the IWC profiles.

Technical corrections:

- Do all the instruments work well all along the 3 months? You should mention it and if there is some missing data. - Line 91: typo : "and was" - Line 175: typo: "summer and winter" - Line 254: error on unities in " 0.25 to 0.75 g.m-3 ". You mean 0.025 to 0.075 ? - Lie 331: Quote the previous studies please. - Line 386: error in unity: " 100g.m-3 ".

Here are some typos in the reference list:

- Line 40: Karlsson and Svensson instead of Karlsson? - Line 100: Crewell and Lohnert instead of Crewell et al? - Line 284: is it Shupe et al (2006) or Shupe (2007) ? - Line 497: Shupe (2011a) not present in the text. - Line 499: 2011b at the end of citation - Line 518: 2014 at

---

## Author Comment (AC2) · 23 Sep 2020

The comment was uploaded in the form of a supplement:
https://acp.copernicus.org/preprints/acp-2020-56/acp-2020-56-AC2-supplement.pdf

---

## Author Comment (AC3) · 23 Sep 2020

The comment was uploaded in the form of a supplement:
https://acp.copernicus.org/preprints/acp-2020-56/acp-2020-56-AC3-supplement.pdf

---

## Author Response (AR1)

**Reviewer 1:**

We would like to thank the Reviewer for the comments and suggestions that helped to improve the quality of our publication. Reviewer comments are given in blue. Our replies are given in black.

*This paper would be much helped by demonstrating a couple of illustrative case studies that show typical events and that illuminate the main points of the paper.*

*For instance, I would like to know what the change in microwave brightness temperature at 31 GHz looks like for a cloud with liquid water path of 10 g m⁻², how such a cloud appears on the w-band radar and whether the lidar is penetrating through the cloud.*

Thank you for this suggestion. We have added Figures 1 and 2 below (Figures 2 and 3 in the revised manuscript). Figure 1 shows a time series taken on 25 July 2014 that illustrates the change in LWP and the raw measurements of brightness temperature at 31.4 GHz as observed at relatively low LWP values during a change from cloudy to cloud-free conditions. The same data is shown in Figure 2 as a scatter plot of LWP and brightness temperature colour coded according to the maximum in radar reflectivity of the respective profile. The Figures show that the 31.4 GHz brightness temperature varies between 15 °C and 16 °C in the absence of cloud. The brightness temperature varies between 17 °C and 18 °C when the retrieval gives a LWP around 10 gm⁻² during the time period from 1100 to 1200 UTC. Visibility is very low until 1400 UTC and the Doppler lidar signal appears to be fully attenuated by the cloud. The cloud produced some precipitation from 1330 UTC and started to disappear around 1400 UTC, when the lidar was able to fully penetrate the cloud.

[Figure]

Figure 1: Time series of cloud radar reflectivity, Doppler lidar backscatter coefficient, LWP, 31.4-GHz brightness temperature, visibility, and rain rate for the time period from 1100 to 1700 UTC on 25 July 2014.

[Figure]

Figure 2: Scatter plot of the change in brightness temperature at 31.4 GHz as measured by the MWR with LWP. Colour coding refers to the maximum in cloud radar reflectivity in dBz for the respective profile. The data cover the time period from 1100 to 1700 UTC on 25 July 2014 as in Figure 1. White circles refer to cloud-free regions.

The figures have been added to Section 2.2. The following text has been added to the manuscript:

*Liquid clouds are diagnosed from lidar and radar profiles. The microwave radiometer provides the LWP associated with these clouds. A time series of LWP and the raw measurements of brightness temperature at 31.4 GHz as observed at relatively low LWP values during a change from cloudy to cloud-free conditions for a single-layer cloud is shown in Figure 2. The brightness temperature varies between 17°C and 18°C when the retrieval gives a LWP around 10 gm$^{-2}$ during the time period from 1100 to 1200 UTC on 25 July 2014. Visibility is very low until 1400 UTC and the Doppler lidar signal appears to be fully attenuated by the cloud. The cloud produced some precipitation from 1330 UTC and started to disappear around 1400 UTC, when the lidar was able to fully penetrate the cloud. The same data is shown in Figure 3 as a scatter plot of LWP and brightness temperature colour coded according to the maximum in radar reflectivity of the respective profile. The figure shows that the 31.4 GHz brightness temperature is lowest at LWP around 0 gm$^{-2}$, i.e. in the absence of cloud, supporting the plausibility of the LWP retrieval and the offset correction.*

*The resolution of the LWP retrieval is about 5 gm$^{-2}$, but the uncertainty in LWP is still of the order of 20-30 gm$^{-2}$ (Turner et al., 2007). The offset correction during cloud-free periods leads to a bias that is much lower than 20 gm$^{-2}$ as shown in Figure 2. The correction is done by using the lidar to diagnose clear sky periods when LWP should be zero and adjust the coefficients for the microwave radiometer retrieval to obtain values around zero. Details are provided in (Gaussiat et al., 2007). Values of LWP below ~25 gm$^{-2}$ in the presence of clouds (as detected from independent measurements) and a known bias (i.e. from offset correction during clear-sky periods) must not be tossed as this leads to a bias in the statistics. Those are still valid values though with an error of ±~25 g m$^{-2}$.*

*Another useful case study would illustrate a cloud system that extends through the inversion and that is precipitating ice versus liquid.*

Examples of two such a cases are shown in Figures 3 and 4 below. The figures have been added to the supplement with a reference to further case studies after the discussion of Figure 2.

[Figure]

Figure 3: Time series of cloud radar reflectivity, Cloudnet target mask, cloud-top temperature, and cloud base and top height for the time period from 0500 to 1300 UTC on 24 July 2014. Crosses and circles mark the location of the main and secondary inversion, respectively.

[Figure]

Figure 4: Time series of cloud radar reflectivity, Cloudnet target mask, cloud-top temperature, and cloud base and top height for the time period from 1400 to 0000 UTC on 25 August 2014. Crosses and circles mark the location of the main and secondary inversion, respectively.

*Precipitation is not addressed at all yet the boundary layer clouds at issue here must precipitate quite often.*

The reviewer is correct that we have neglected the issue of raining clouds. We have now looked at the occurrence of precipitation (drizzle, rain, and ice below liquid clouds) in the Cloudnet target mask to better address the precipitation issue. We have added two new cloud class called "precipitating liquid clouds" and "precipitating mixed-phase clouds" to our analysis. The definition of these classes has been added to the description in Section 2.2. The Figures affected by this change and their

discussion have been revised accordingly. We find that precipitating clouds are more abundant during summer. This is now displayed in Figure 5 (former Figure 3). Figure 4 (former Figure 2) shows that precipitating clouds are linked to frontal passages, i.e. deep cloud systems. We have encountered more of those during summer while stable boundary layers with shallow stratus clouds, which are typical for observations in the marginal ice zone, prevailed during autumn. A corresponding statement has been added to the discussion of Figure 5 (former Figure 3):

*"Figure 5 also reveals that precipitating clouds were more abundant during summer than during winter. This is in line with Figure 4 which shows that precipitating clouds are linked to frontal passages, i.e. deep cloud systems. More of such deep cloud systems have been encountered during summer. In contrast, a stable boundary layer with shallow stratus clouds, which typical occur in the marginal ice zone, prevailed during autumn."*

We have also separated the presentation of cloud statistics (Figures 4, 7, 8, and 10) of the original submission into precipitating and non-precipitating clouds. The discussion has been extended accordingly. The revised figures look like this:

[Figure]

Figure 5: Profiles of cloud fraction for different cloud types as obtained using Cloudnet for the entire ACSE campaign (left), summer (middle), and autumn (right). All solid profiles refer to clouds for which the cloud-top height was located below 6 km height. The dashed lines refer to the total cloud fraction with respect to all clouds, i.e. including those with undetected cloud-top heights.

[Figure]

Figure 6: Statistical overview of cloud occurrence with respect to (a) top temperature, (b) top height, (c) base height, and (d) geometrical thickness for the entire ACSE campaign (left column) as well as for summer (middle column) and autumn (right column). The colours indicate the different cloud types as in Figure 3: all (black), non-precipitating liquid (red), precipitating liquid (magenta), non-precipitating mixed-phase (dark green), precipitating mixed-phase (light green), and ice (blue). The numbers in the top panel refer to the number of 30-s profiles considered in the analysis.

[Figure]

Figure 7: Statistics on the geometrical thickness (a and c) and the frequency distribution of cloud-top-temperature (b and d) of non-precipitating (a and b) and precipitating liquid clouds (c and d) clouds observed for different PBL structure, temperature in the free troposphere, and location with respect to the main inversion: non-WAE with coupled PBL (blue), non-WAE with decoupled PBL (red), and WAE with coupled PBL (green). The different boxes (a and b) and lines in (c and d) refer to clouds with cloud base above the inversion (above), to clouds with cloud top below the inversion (below) or to clouds with cloud base below the inversion and cloud top above the inversion (through). Numbers in (a) and (b) refer to the number of Cloudnet profiles per category. Categories with less than 100 profiles have been omitted; this includes all cases of decoupled PBL during WAE.

[Figure]

Figure 8: Same as Figure 7 but for non-precipitating (a and b) and precipitating mixed-phase clouds (c and d).

[Figure]

Figure 9: Scaled profiles of LWC for non-precipitating (a) and precipitating liquid clouds (b) observed for different PBL structure, temperature in the free troposphere, and location with respect to the main inversion. Zero and unity of the scaled altitude refer to cloud base and top, respectively. Categories with less than 100 profiles have not been included.

[Figure]

Figure 10: Scaled profiles of LWC (a and b) and IWC (c and d) for non-precipitating (a and c) and precipitating mixed-phase clouds (b and d) observed for different PBL structure, temperature in the free troposphere, and location with respect to the main inversion. Zero and unity of the scaled altitude refer to cloud base and top, respectively. Categories with less than 100 profiles have not been included.

[Figure]

Figure 11: Fraction of non-precipitating (solid) and precipitating (dashed) mixed-phase clouds observed during ACSE in summer (black) and autumn (red) in comparison to the entire ACSE data set (gray) as well as to previous observations of mid-level clouds at mid-latitude from ground (Bühl et al., 2013) and space (Zhang et al., 2010).

*While the occurrence of semi-transparent liquid clouds in the Arctic have been known for some time, these are normally diagnosed with an infrared spectrometer and ceilometer. Is there something special about the microwave radiometer used in this study and the algorithm used in the inversions that allow it to have precision below the 20 or 30 g m⁻² that is typical of such instruments? This seems to be the level of precision cited in the referenced papers. The authors describe an "offset correction" for the LWP based on clear sky periods. What does this mean and how is determined and justified?*

Liquid clouds are diagnosed from ceilometer and radar profiles, the microwave radiometer provides the LWP associated with these clouds. The resolution of the LWP retrieval is about 5 g m$^{-2}$, but the uncertainty in LWP is still of the order of 20-30 g m$^{-2}$. Values of LWP below ~25 g m$^{-2}$ in the presence of clouds (as detected from independent measurements with e.g. cloud radar, ceilometer, or Doppler lidar) must not be tossed as this leads to a bias in the statistics. Those are still valid values with an error of ±~25 g m$^{-2}$. The noise level during cloud-free periods is much lower than 20 g m$^{-2}$ as shown in Figure 1 above.

The offset correction for LWP during clear-sky periods does not affect the cloud statistics. The correction is done by using the Doppler lidar or ceilometer to diagnose clear sky periods when LWP should be zero and adjust the coefficients for the microwave radiometer retrieval to obtain values around zero. Details are provided in Gaussiat et al. (2007).

Gaussiat, N., R. J. Hogan, and A. J. Illingworth, 2007: Accurate Liquid Water Path Retrieval from Low-Cost Microwave Radiometers Using Additional Information from a Lidar Ceilometer and Operational Forecast Models. J. Atmos. Oceanic Technol., 24, 1562–1575, https://doi.org/10.1175/JTECH2053.1.

A similar approach is also used for the ARM MWWRET retrieval (Turner et al., 2007).

Turner, D. D., S. A. Clough, J. C. Liljegren, E. E. Clothiaux, K. E. Cady-Pereira and K. L. Gaustad, Retrieving Liquid Water Path and Precipitable Water Vapor From the Atmospheric Radiation Measurement (ARM) Microwave Radiometers, in IEEE Transactions on Geoscience and Remote Sensing, vol. 45, no. 11, pp. 3680-3690, Nov. 2007, https://doi.org/10.1109/TGRS.2007.903703.

This information has been added to the new text in the description of the MWR retrieval (given above).

*The authors use an algorithm to categorize profiles that was published in 2007 as part of the Cloudnet program. I am sceptical that this algorithm could be used reliably without modification from one data set to another since such algorithms always rely on various tunable thresholds to make decisions. I wonder if the authors could describe the thresholds used by this algorithm to classify profiles between liquid, mixed, and ice and whether they found it necessary to adjust the thresholds in the algorithm.*

The Cloudnet algorithm was designed to be used across multiple locations without tunable parameters - i.e. an objective classification - even for sites with different combinations of instruments. In practice, the only parameter requiring 'tuning' is for insect discrimination, which, with most cloud radars now capable of providing LDR, also more objective. Liquid cloud classification uses the ceilometer attenuated backscatter profile and the cloud base detection is a combination of threshold (maximum value in the profile) and profile shape. There is no tuning of these parameters from site to site, and it is the profile shape which is of most importance when diagnosing liquid layers; sensitivity to the choice of threshold value is quite low. Ice is classified using the radar reflectivity and fall velocity, so together with the liquid

diagnosis from the ceilometer, one pixel may be classified as liquid, ice, or both. The algorithm parameters are not varied from site to site.

We have added a statement to the description of the Cloudnet retrieval regarding the effect of instrumental uncertainties:

The measurement uncertainties of the individual instruments are used to obtain a data quality flag. This study only considers profiles that are flagged as reliable and show a standard deviation of the LWP smaller than 120 gm$^{-2}$.

We have also revised the description of our definition of the different cloud phases for more clarity. The text now states:

*"A cloud is defined as liquid if its profile contains only height bins that are classified as Cloud droplets only or Drizzle/rain & cloud droplets. A cloud for which all height bins are classified as Ice only is defined as ice cloud. A cloud layer is defined as mixed-phase, if it contains any possible combination of Ice only, Cloud droplets only, Melting ice, Melting ice & cloud droplets, and Ice & super-cooled droplets. Finally, layers of Cloud droplets only with precipitating ice below cloud base and mixed-phase clouds with Drizzle or rain below cloud base are defined as precipitating mixed phase. Liquid clouds with liquid precipitation are defined as precipitating liquid. Profiles of cloud fraction per volume (Brooks et al., 2005) have been obtained using time-height sections of 30 min by 90 m height (3 height bins). When comparing our findings to results from previous studies that use the cloud classification of Shupe (2007), we sort all layers of Cloud droplets only with precipitating ice below cloud base into the mixed-phase category to be in line with the earlier work."*

*The Hogan et al (2006) IWC algorithm uses airborne data collected over the UK in mostly frontal clouds as I recall using a Mass-Size relationship developed by Brown and Francis in such clouds. Is there a reason to suspect that the ice precipitation produced in the Arctic boundary layer is similar to frontal ice precipitation over the UK so that the same regression algorithm could be used here?*

It is true that the Hogan et al (2006) IWC-Z-T algorithm is based on aircraft observations in mid-latitude ice clouds, however, the algorithm compares well with other algorithm-intercomparisons for Cloudsat observations spanning the whole globe (*Stein et al.*, 2011)

Stein, T. H. M., J. Delanoë, and R. J. Hogan, 2011: A Comparison among Four Different Retrieval Methods for Ice-Cloud Properties Using Data from CloudSat, CALIPSO, and MODIS. J. Appl. Meteor. Climatol., 50, 1952–1969, https://doi.org/10.1175/2011JAMC2646.1.

*Perhaps I missed it but how often were radiosondes launched during this voyage?*

Radiosondes were launched in 6-h intervals. This has now been clarified in various locations in the text. The exact launch times were given in Section 2.2.

*A binary classification of warm air advection based on the temperature at 1 km seems problematic to me. Warm air advection implies large scale transport with the wind perpendicular to a thermal gradient. True warm air advection would have real implications for clouds because it is typically accompanied by deep large-scale*

*ascent, etc. Perhaps the authors could show statistics of the strength of the advection in units of temperature per unit time and how the cloud distributions respond to this?*

The Referee is correct that our reference to advection is not ideal. We have changed the description to clarify that we are referring to situations during which warm air was present in the free troposphere (warm air events, WAE) and during which it was not (non-warm air events, non-WAE) following the criterion provided in the text. We have dropped the term warm-air advection and the reference to the large-scale circulation from the discussion. The text, the figures, and the discussion have been revised accordingly.

*"We further separate between conditions during which warm air was present in the free troposphere (warm air events, WAE) and during which it was not (non-warm air events, non-WAE). WAE were identified from the ACSE soundings as when the temperature at 1.0 km height exceeded a threshold of 5 C, empirically derived from Figure 2a of Sotiropoulou et el. (2016). These events were particularly pronounced during the ACSE summer observations and are likely the result of warm-air advection from lower latitudes (Tjernström et al., 2015; 2019)."*

*The vertical structure of mixed phase clouds is important. In these Arctic clouds typically ice precipitation sediments from a thin liquid water layer. Is that what is seen here? Is the radar able to detect the water layers? If so, how often? Is the MWR capable of retrieving the water path from such layers? If so, what is the typical increase in microwave brightness temperature? A case study would be interesting and useful to establish confidence in the analysis methodology.*

The occurrence of liquid-water clouds that precipitate ice is shown in the Cloudnet target mask in Figure 4 (former Figure 2). Cloudnet can identify such cases and they are classified as mixed-phased cloud in our study. We show in new Figure 12 (Figure 10 above) that LWP and IWP are retrieved for these clouds.

*There is no mention of drizzle, light rain or snow. I'm not sure how a study of clouds in the Arctic could be complete without considering the properties of the precipitation that certainly occurs much of the time.*

Please see our reply to the earlier comment on rain above. We had no precipitation sensor that would allow us to separate between rain and snow. We have now added two new cloud class called "precipitating liquid clouds" and "precipitating mixed-phase clouds" to our analysis. The definition of these classes has been added to the description in Section 2.2. The Figures affected by this change and their discussion have been revised accordingly.

**Reviewer 2:**

*There were no in-situ observations of these clouds, and so further discussion of measurement uncertainties or else comparison to in-situ aircraft observations of similar clouds in the Arctic or mid latitudes would be welcome.*

We have thought about a comparison to in-situ measurements from the literature but found it hard to come up with a meaningful way of doing so. This is related to the location and time of the measurements as well as the differences in sampling techniques. Most importantly, however, it is impossible to ensure that clouds probed during in-situ observations are in any way comparable to the clouds we probed during our observations. We therefore limit the comparison to satellite observations, for which comparable measurement techniques and data analysis methodologies are used. Nevertheless, we have produced a comparison to the in-situ profiles of cloud temperature, LWC, and IWC for single-layer mixed-phase clouds presented in *Mioche et al.* (2017). We find similar shapes of the LWC and IWC profiles compared to their findings, which are based on in-situ aircraft observations. A corresponding Figure S3 has been added to the Supplementary Material will a reference in the text during the discussion of the LWC and IWC profiles.

Mioche, G., Jourdan, O., Delanoë, J., Gourbeyre, C., Febvre, G., Dupuy, R., Monier, M., Szczap, F., Schwarzenboeck, A., and Gayet, J.-F.: Vertical distribution of microphysical properties of Arctic springtime low-level mixed-phase clouds over the Greenland and Norwegian seas, Atmos. Chem. Phys., 17, 12845–12869, https://doi.org/10.5194/acp-17-12845-2017, 2017.

[Figure]

Figure S3: Scaled in-cloud profiles of temperature, LWC, and IWC during summer (solid lines) and autumn (dashed lines) for single-layer mixed phase clouds sorted into the warm and cold categories analogous to *Mioche et al.* (2017).

*The authors also might consider adding more speculations about the physical mechanisms behind the observations. For instance, why are the clouds mixed-phase at such warm temperatures (Fig. 10)? More physical explanation might help better tie together the observations into a more coherent understanding.*

The clouds with warm cloud-top temperatures are those that extend through the inversion into the free troposphere during conditions in which we observed warmer air aloft. We now separate between precipitating and non-precipitating clouds in Figure 14 (old Figure 10) and find that the mixed-phase clouds with the warmest cloud-top temperatures are also precipitating clouds and that they are already fully glaciated at a top-temperature of -10°C.

We have added a case study to the supplementary material that illustrates the occurrence of clouds that extend through the inversion. We have also added Supplementary Material that presents two detailed examples of the connections between cloud extent, inversion height, and cloud-top temperature.

*Specific comments:*

*Lines 116–118: "Liquid water content (LWC) is calculated from microwave radiometer derived LWP (with an offset correction based on clear-sky periods) and liquid layer cloud boundaries by distributing the liquid using the scaled-linear adiabatic assumption, i.e. LWC increasing linearly with height from zero at cloud base." So LWC is assumed to increase linearly with height, but then in Fig. 8, the LWC is plotted and the authors conclude "Liquid clouds show maximum LWC between 0.03 and 2.00 gm−3 within the upper quarter of the cloud". Is this a conclusion about nature or merely an artefact of the authors' assumption of linearly increasing LWC with height?*

The Referee is raising an important issue. Our profiles of LWC presented in Figures 11, 12, and S3 show the same shape as found in the in-situ measurements presented in Figure 4c of *Mioche et al.* (2017). We therefore believe that this observation is no artefact. A corresponding statement has been added to the text.

*Lines 143–146: "We further separate the large-scale circulation between warm-air advection events (WAA, Tjernström et al. (2015) and conditions during which no warm-air advection took place (non-WAA). WAA was identified from the ACSE soundings as when the temperature at 1.0 km height exceeded a threshold of 5 C, empirically derived from Figure 2a of Sotiropoulou et al.(2016)." Is this simple temperature criterion sufficient to accurately diagnose warm-air advection? Can it be corroborated by, e.g., wind observations showing wind from southern, warmer regions?*

The Referee is correct that our reference to advection is not ideal. We have therefore changed the description to clarify that we are referring to situations during which warm air was present in the free troposphere (warm air events, WAE) and during which it was not (non-warm air events, non-WAE) following the criterion provided in the text. We have dropped the term warm-air advection and the reference to the large-scale circulation from the discussion. The text, the figures, and the discussion have been revised accordingly.

*"We further separate between conditions during which warm air was present in the free troposphere (warm air events, WAE) and during which it was not (non-warm air events, non-WAE). WAE were identified from the ACSE soundings as when the temperature at 1.0 km height exceeded a threshold of 5 C, empirically derived from Figure 2a of Sotiropoulou et el. (2016). These events were particularly pronounced*

*during the ACSE summer observations and are likely the result of warm-air advection from lower latitudes (Tjernström et al., 2015; 2019)."*

*Lines 247–248: "Figure 8 provides a profile view of the LWC and the IWC of the clouds considered in Figure 7. The scaled altitude ranges from the base of the clouds (zero) to the cloud top (unity)." In this figure, is the cloud base defined as liquid cloud base, ice cloud base, or the lower of the two? In the leftmost (liquid) panel, some clouds exhibit non-zero liquid at cloud base, which seems contradictory. Shouldn't cloud base in a liquid-only cloud be defined as the altitude at which liquid goes to zero? If so, how can the liquid remain non-zero at cloud base? In the middle and right-hand (mixed-phase) panels, all but two of the cloud layers have liquid extending all the way to cloud base, and ice approaching zero at cloud base. Why weren't there more clouds with ice falling out of liquid cloud base? Is this an observational artefact?*

The cloud base refers to the liquid cloud base. LWC larger than zero at cloud base can occur when the transition between cloud and cloud-free does not align with the coarse binning of the remote-sensing instruments.

*Line 329–330: "While they find that about 50% of all clouds are mixed-phase at a temperature of about -10 C, the ACSE observations reveal that in the Arctic a mixed-phase cloud fraction of 50% is reached already at -2 C." Can the authors speculate about why this difference is observed in these datasets for Arctic and mid-latitude clouds? E.g., do more of the Arctic clouds have cold regions above that nucleate ice, which then sediments down to warmer temperatures (e.g., Fleishauer et al. 2002)?*

The most likely reason is the occurrence of ice-crystal seeding from above and secondary ice formation in the clouds. The following text has been added to the description of Figure 14 (former Figure 10):

*Because Arctic clouds often occur in the form of multi-layered clouds, it is most likely that ice-crystal seeding from upper-level clouds into lower-level clouds leads to the high mixed-phase cloud fraction at relatively high temperatures (Vassel et al., 2019).*

*Lines 350–351: "The ACSE data set reveals a strong reduction in the occurrence rate of liquid clouds and an increase for both mixed-phase clouds and ice clouds at low levels during autumn compared to summer." To me, a striking difference between summer and autumn is that there is more drizzle/rain during summer (see Fig. 2). I encourage the authors to comment on this if they have a hypothesis why this is observed.*

The Referee is correct. We have added precipitating clouds to the analysis and find that they are more abundant during summer. This is now displayed in Figure 5 (former Figure 3). Figure 4 (former Figure 2) shows that precipitating clouds are linked to frontal passages, i.e. deep cloud systems. We have encountered more of those during summer while stable boundary layers with shallow stratus clouds, which are typical for observations in the marginal ice zone, prevailed during autumn. A corresponding statement has been added to the discussion of Figure 5 (former Figure 3):

*"Figure 5 also reveals that precipitating clouds were more abundant during summer than during winter. This is in line with Figure 4 which shows that precipitating clouds*

*are linked to frontal passages, i.e. deep cloud systems. More of such deep cloud systems have been encountered during summer. In contrast, a stable boundary layer with shallow stratus clouds, which typical occur in the marginal ice zone, prevailed during autumn."*

We have also separated the presentation of cloud statistics (Figures 4, 7, 8, and 10) of the original submission into precipitating and non-precipitating clouds. The discussion has been extended accordingly. The revised figures are shown in the reply to Referee 1.

*Fig. 1: Is it possible to display more clearly the area covered by sea ice using, e.g., light-blue shading?*

We did not change the figure. Sea-ice extend is shown for two time periods. Showing those as areas would distract too much from other information.

*Fig. 2: The red-green-blue (liquid-mixed-ice) colour convention of Figures 3 and 4 is not followed in Fig. 2, which uses yellow to denote ice, e.g.. Would it be clearer to make the colour convention of Fig. 2 match that of Figures 3 and 4?*

The reviewer is correct. Ice is shown in yellow in the Cloudnet cloud mask. However, the use of yellow in the other plots would not be practical as those lines would be very hard to see.

*Fig. 2 seems to suggest that ice clouds are geometrically thicker than mixed-phase clouds, whereas Fig. 4 seems to suggest that it's the other way around. Which is it? Are these two figures consistent with each other?*

Thank you for this comment. The two figures are indeed consistent. We would like to point out that any yellow lines in Figure 2 that are broken by a green region (supercooled droplets) are considered as mixed-phase in line with our definition provided in the description of the Cloudnet retrieval in Section 2.2. Nevertheless, we have revised the description of our cloud-phase definition in Section 2.2 to:

*"A cloud is defined as liquid if its profile contains only height bins that are classified as Cloud droplets only or Drizzle/rain & cloud droplets. A cloud for which all height bins are classified as Ice only is defined as ice cloud. A cloud layer is defined as mixed-phase, if it contains any possible combination of Ice only, Cloud droplets only, Melting ice, Melting ice & cloud droplets, and Ice & super-cooled droplets. Finally, layers of Cloud droplets only with precipitating ice below cloud base and mixed-phase clouds with Drizzle or rain below cloud base are defined as precipitating mixed phase. Liquid clouds with liquid precipitation are defined as precipitating liquid. Profiles of cloud fraction per volume (Brooks et al., 2005) have been obtained using time-height sections of 30 min by 90 m height (3 height bins). When comparing our findings to results from previous studies that use the cloud classification of Shupe (2007), we sort all layers of Cloud droplets only with precipitating ice below cloud base into the mixed-phase category to be in line with the earlier work."*

*Reference:*

*Fleishauer, R. P., Larson, V. E., & Vonder Haar, T. H. (2002). Observed microphysical structure of midlevel, mixed-phase clouds. Journal of the atmospheric sciences, 59(11), 1779-1804.*

***Reviewer 3:***

*My main comments are below:*

*- My first main remark is that the authors could greatly enhance the discussion of the results by including the environmental conditions when comparing the cloud properties (and not limit only to the comparison between summer and autumn). Indeed, I think the paper would be greatly improved by a section presenting the meteorological conditions during ACSE. Not a detailed characterization of course, but at least some statistics about environmental parameters (temperature, humidity, winds, air mass origins) in order to identify the main regimes encountered during the experiment. In this new section, you may include for example the description of the WAA events.*

We agree with the reviewer that a brief discussion of the meteorological conditions is useful for the paper. Instead of adding new plots, however, we are referring to earlier publications where environmental parameters are discussed in more detail. We have added the following text to Section 2.1:

*"In contrast to the majority of shipborne cloud observations in the Arctic, ACSE measurements were performed when the ship was moving. Hence, the measurements were taken over open water as well as partial or complete sea ice cover, and the ice appeared with and without melt ponds and snow cover. The moving platform complicates a statistical analysis of the meteorological situation and we provide only a basic overview here. Meteorological instruments and the measured conditions are presented in more detail in Sotiropoulou et al. (2016) and Sedlar et al. (2020). The impact of meridional heat transport on the surface energy budget during ACSE is described in Tjernström et al. (2015, 2019).*

*Sotiropoulou et al. (2016) used the lower atmospheric thermal structure as inferred from 6-hourly soundings to separate the ACSE period into two seasons (see their Figure 2). Before 1200 UTC on 27 August 2014, relatively high temperatures prevailed in the lower troposphere up to 5 km height, with occasional cooler periods. Several strong warm-air advection events occurred during this first half of the experiment, which Sotiropoulou et al. (2016) refer to as the summer melt season, the strongest is described in Tjernström et al. (2015). After 27 August 2014 the lowermost kilometres of the atmosphere changed substantially with a decreased temperature, with only a few occasional warmer events, considered to represent autumn freeze-up (about 42% of the ACSE time period). Figure 2b in Sotiropoulou et al. (2016) shows the temperature at the main inversion, i.e. the strongest inversion in the radiosonde profiles used as a proxy for the top of the boundary layer. This is mostly positive for the summer period and generally negative during the autumn period.*

*Figure 2 in Sedlar et al. (2020) present time series of select near-surface meteorological parameters, indicating a number of synoptic weather events that were encountered along the ACSE track. These events occurred more often in the second half of the experiment, though with a shorter duration. Surface pressure minima first dropped below 1,000 hPa on 27 August, the same date that Sotiropoulou et al. (2016) defined as the seasonal transition from summer to autumn. Near-surface wind*

*speed also peaks more often and slightly higher after this date, compared to earlier. The transition is also visible in near-surface temperature, which remained at or above freezing level before 27 August then fell down to, or below, the freezing point.*

*Figure 2 shows that fog occurred much more frequently during the summer melt season compared to the autumn freeze-up. The difference in cloud occurrence and properties between the two seasons is discussed in detail later in this paper.*

*- Moreover, since all the results are based on remote sensing measurements, I think the authors could develop the uncertainties and limitations of such instrumentation in more details.*

Thank you for this comment. The following text has been added to the description of the Cloudnet retrieval to resolve the issue:

*The measurement uncertainties of the individual instruments are used to obtain a data quality flag. This study only considers profiles that are flagged as reliable and show a standard deviation of the LWP smaller than 120 gm$^{-2}$.*

*- You compare your results with satellite observations, but not to in situ measurements. I would suggest to do it, keeping in mind that in situ measurements remain very localized in time and space.*

We have thought about a comparison to in-situ measurements from the literature but found it hard to come up with a meaningful way of doing so. This is related to the location and time of the measurements as well as the differences in sampling techniques. Most importantly, however, it is impossible to ensure that clouds probed during in-situ observations are in any way comparable to the clouds we probed during our observations. We therefore limit the comparison to satellite observations, for which comparable measurement techniques and data analysis methodologies are used. Nevertheless, we have produced a comparison to the in-situ profiles of cloud temperature, LWC, and IWC for single-layer mixed-phase clouds presented in *Mioche et al.* (2017). We find similar shapes of the LWC and IWC profiles compared to their findings, which are based on in-situ aircraft observations. A corresponding Figure S3 has been added to the Supplementary Material will a reference in the text during the discussion of the LWC and IWC profiles.

Mioche, G., Jourdan, O., Delanoë, J., Gourbeyre, C., Febvre, G., Dupuy, R., Monier, M., Szczap, F., Schwarzenboeck, A., and Gayet, J.-F.: Vertical distribution of microphysical properties of Arctic springtime low-level mixed-phase clouds over the Greenland and Norwegian seas, Atmos. Chem. Phys., 17, 12845–12869, https://doi.org/10.5194/acp-17-12845-2017, 2017.

[Figure]

Figure S3: Scaled in-cloud profiles of temperature, LWC, and IWC during summer (solid lines) and autumn (dashed lines) for single-layer mixed phase clouds sorted into the warm and cold categories analogous to *Mioche et al.* (2017).

*- Line 96: Please, could you explain how is used the training dataset in the radiometer retrieval algorithm and why you could not use the radiosondes from ACSE? Moreover, in this section 2.2, you mention the radiosondes and the vertical profiles of temperature and humidity made during ACSE. I suggest the authors to show some statistics/figures about these data to present the context of your results (you may include them in the "meteorological conditions", cf. previous comment).*

We have compiled a training data set that can considered as climatologically meaningful in relation to the ACSE observations. ACSE radiosondes have been included in this training data set. In general, the result of the training improves with the size of the training data set. We have added a statement to clarify that ACSE radiosonde observations are included in the training data set.

*- The presented statistics are made for each identified cloud layer. Since you analyse up to three by profiles (cf. line 166), I think it could be interesting to calculate the ratio of single layer cloud vs multilayer cloud (with keeping in mind this maximum limit of 3 layers).*

We only use the remote-sensing observations and the Cloudnet target classification to identify cloud layers. This gives us apparent cloud layers and apparent multi-layer clouds for which features have to be clearly separated in a profile. During summer, we find occurrence rates of 19.6% cloud-free conditions, 39.1% single-layer clouds, and 41.3% multi-layer clouds. During autumn, these numbers change to 4.6%, 47.6%, and 47.8%. This means that apparent single-layer clouds and multi-layer clouds occur at about the same rate during both seasons.

*- In lines 122 to 131 you present the target classification. The readability would be enhanced with a schemed summarizing this classification.*

Cloudnet is an established retrieval methodology. The Cloudnet target classification is discussed in detail in *Illingworth et al.* (2007). The targets of relevance for this study are mentioned in the text and the full time series if the target mask is shown in Figure 2 (now Figure 4). We belief that providing the reference for further reading is sufficient for readers that would like to know specific details of the Cloudnet algorithm.

*- Was there an instrument on board to measure the precipitation? It could be of great interest to identify the precipitating clouds from the non-precipitating. Precipitation is closely linked to cloud properties and is a key factor but still difficult to assess. Also, it would help the discussion, for example in lines 373-374 on the IWC profiles.*

The reviewer is correct that we have neglected the issue of raining clouds. This is because we found assessing the occurrence of rain during ACSE is not straightforward. The rain sensor on the ship could not be the first choice for such an investigation as its measurements were strongly affected by persistent fog as well as sea spray produced by the icebreaker itself. We have looked at the occurrence of precipitation (drizzle, rain, and ice below liquid clouds) in the Cloudnet target mask to better address the precipitation issue. We have now added two new cloud class called "precipitating liquid clouds" and "precipitating mixed-phase clouds" to our analysis. The definition of these classes has been added to the description in Section 2.2. The Figures affected by this change and their discussion have been revised accordingly. We find that precipitating clouds are more abundant during summer. This is now displayed in Figure 5 (former Figure 3). Figure 4 (former Figure 2) shows that precipitating clouds are linked to frontal passages, i.e. deep cloud systems. We have encountered more of those during summer while stable boundary layers with shallow stratus clouds, which are typical for observations in the marginal ice zone, prevailed during autumn. A corresponding statement has been added to the discussion of Figure 5 (former Figure 3):

*"Figure 5 also reveals that precipitating clouds were more abundant during summer than during winter. This is in line with Figure 4 which shows that precipitating clouds are linked to frontal passages, i.e. deep cloud systems. More of such deep cloud systems have been encountered during summer. In contrast, a stable boundary layer with shallow stratus clouds, which typical occur in the marginal ice zone, prevailed during autumn."*

We have also separated the presentation of cloud statistics (Figures 4, 7, 8, and 10) of the original submission into precipitating and non-precipitating clouds. The discussion has been extended accordingly. The revised figures are shown in the reply to Referee 1.

*- Do all the instruments work well all along the 3 months? You should mention it and if there is some missing data.*

Periods of instrument downtime for which no Cloudnet retrieval could be performed are marked as hatched areas in Figure 2 (now Figure 4) as stated in the figure

caption: "*Hatched areas separate periods of no data from the white background of Clear sky.*"

*- Line 91: typo : "and was"*

changed

*- Line 175: typo: "summer and winter"*

The statement was changed to: "…depths of 750 m in summer and 940 m in winter, with a similar mean value…"

*- Line 254: error on unities in " 0.25 to 0.75 g.m-3 ". You mean 0.025 to 0.075?*

Indeed, changed.

*- Lie 331: Quote the previous studies please.*

These studies are discussed in this paragraph. For clarity, the beginning of the second sentence was changed to: "*The available data sets discussed below are...*"

*- Line 386: error in unity: " 100g.m-3 ".*

This is no error. We are referring to the most extreme values in the data set.

*Here are some typos in the reference list:*

All typos have been revised

*- Line 40: Karlsson and Svensson instead of Karlsson?*

*- Line 100: Crewell and Lohnert instead of Crewell et al?*

*- Line 284: is it Shupe et al (2006) or Shupe (2007) ?*

It's Shupe (2007), as stated.

*- Line 497: Shupe (2011a) not present in the text.*

*- Line 499: 2011b at the end of citation*

*-Line 518: 2014 at*

This reference was changed to:

[revised manuscript text omitted]

This supplementary material presents two case studies (Figures S1 and S2) regarding situations in which a cloud system (i) extends through the inversion and (ii) precipitates ice versus liquid water. The figures display the cloud radar reflectivity together with the height of the main and secondary inversions, the Cloudnet target mask, and the cloud-top temperature as well as the classification of a cloud profile into mixed, liquid, or ice.

We also present scaled in-cloud profiles of temperature, LWC, and IWC for warm and cold single-layer mixed phase clouds during summer and autumn (Figure S3) analogous to the in-situ aircraft profiles in Figures 3, 4c, and 5c in *Mioche et al.* (2017). We followed the description in *Mioche et al.* (2017) to focus on single-layer mixed phase clouds. The observations were separated into clouds with a top temperature in the range from -8°C to -15°C (warm clouds) and with a top temperature in the range from -15°C to -22°C (cold clouds). Note that the airborne in-situ measurements shown in *Mioche et al.* (2017) took place in March, April, and May around the Svalbard archipelago while ACSE observations cover July, August, and September as well as a different region.

Mioche, G., Jourdan, O., Delanoë, J., Gourbeyre, C., Febvre, G., Dupuy, R., Monier, M., Szczap, F., Schwarzenboeck, A., and Gayet, J.-F.: Vertical distribution of microphysical properties of Arctic springtime low-level mixed-phase clouds over the Greenland and Norwegian seas, Atmos. Chem. Phys., 17, 12845–12869, https://doi.org/10.5194/acp-17-12845-2017, 2017.

**Case study of a cloud system that extends through the inversion**

[Figure]

Figure S1: Time series of cloud radar reflectivity, Cloudnet target mask, cloud-top temperature, and cloud base and top height for the time period from 0500 to 1300 UTC on 24 July 2014. Crosses and circles mark the location of the main and secondary inversion, respectively.

**Case study of a cloud system that is precipitating ice versus liquid**

[Figure]

Figure S2: Time series of cloud radar reflectivity, Cloudnet target mask, cloud-top temperature, and cloud base and top height for the time period from 1400 to 0000 UTC on 25 August 2014. Crosses and circles mark the location of the main and secondary inversion, respectively.

**Comparison to scaled profiles in Mioche et al. (2017)**

[Figure]

Figure S3: Scaled in-cloud profiles of temperature, LWC, and IWC during summer (solid lines) and autumn (dashed lines) for single-layer mixed-phase clouds sorted into the warm and cold categories analogous to *Mioche et al.* (2017).